# Boosting Monocular Metric Depth Estimation via Bokeh Rendering

**Hangwei Zhang** [1 2]  **Armando Fortes** [1]  **Tianyi Wei** [1]  **Xingang Pan** [1]

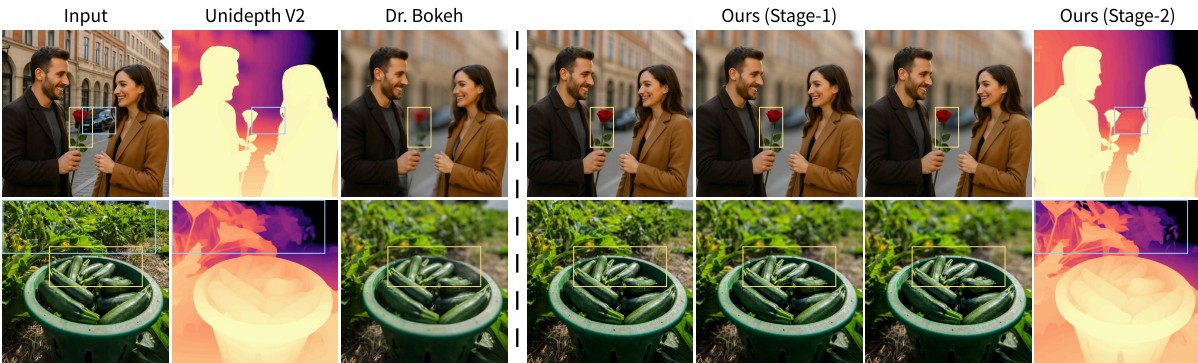

*Figure 1.* **BokehDepth** decouples bokeh synthesis from depth prediction and uses lens-aware defocus as a supervision-free geometric cue to improve the accuracy and physical consistency of monocular depth estimation. *Left:* conventional pipelines predict depth from a single sharp image and render bokeh from the noisy depth map. *Right:* our two-stage framework, where Stage-1 generates a calibrated bokeh stack from a single image and Stage-2 built on UniDepthV2 (Piccinelli et al., 2025) fuses defocus cues to produce sharper and more reliable metric depth.

## Abstract

Bokeh rendering and depth estimation share a fundamental optical connection, yet existing methods fail to fully exploit this reciprocity. Conventional bokeh pipelines rely heavily on noisy depth maps that inevitably introduce visual artifacts. Conversely, existing monocular depth models typically follow two flawed paradigms. Generative diffusion-based frameworks often lack consistent metric scale. Meanwhile, feed-forward metric depth models frequently fail in textureless or distant regions where defocus blur can provide geometric information. We propose **BokehDepth**, a two-stage framework that treats synthetic defocus as a supervision-free geometric signal. In the first stage, a physically grounded generative model produces calibrated bokeh stacks from a single sharp input without requiring prior depth input. Subsequently, a lightweight defocus-aware aggregation module integrates these stacks into the encoder of a depth estimation framework. This mechanism allows the model to extract consistent geometric features from the defocus dimension while keeping the decoder architecture unchanged. Experiments demonstrate that **BokehDepth** achieves superior visual bokeh fidelity compared to depth-dependent rendering baselines and consistently enhances the metric accuracy of state-of-the-art monocular depth models. Project page: https://fogradio.github.io/BokehDepth_Project/.

## 1. Introduction

Bokeh is a lens-originated optical effect that describes the aesthetic quality of out-of-focus regions, which often helps emphasize in-focus subjects by softly blurring distracting details (Mandl et al., 2024). monocular metric depth estimation aims to predict the depth of each pixel relative to the camera from a single RGB image to recover the scene's 3D geometry with scale (Saxena et al., 2008; Eigen et al., 2014; Ranftl et al., 2020). monocular metric depth estimation and bokeh synthesis are intrinsically linked by the lens imaging geometry (Pentland, 1987; Subbarao & Surya, 1994). Accurate depth maps enable physically consistent and controllable bokeh rendering (Sheng et al., 2024; Luo et al., 2023), and defocus cues from bokeh supply informative signals that help resolve geometric ambiguities in depth prediction (Tang et al., 2017; Wijayasingha et al., 2024).

Most high-quality bokeh rendering pipelines still rely on a depth or disparity map to guide spatially varying blur (Peng et al., 2022a;b). This requirement increases system complexity and makes the final visual quality tightly bounded by the depth estimator (Seizinger et al., 2025). Any local depth error is immediately translated into an incorrect blur radius or a broken occlusion edge (Zhu et al., 2025). Classical

[1]S-Lab, Nanyang Technological University [2]Beihang University. Correspondence to: Xingang Pan <xingang.pan@ntu.edu.sg>.

*Proceedings of the 43rd International Conference on Machine Learning*, Seoul, South Korea. PMLR 306, 2026. Copyright 2026 by the author(s).

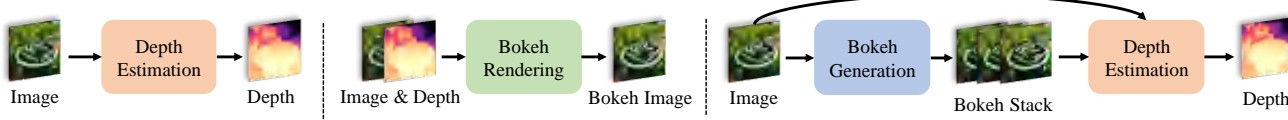

(a) Monocular Depth Estimation     (b) Traditional Bokeh Rendering     (c) Overview of *BokehDepth*

*Figure 2.* From monocular depth and depth-based bokeh to *BokehDepth*. (a) Standard monocular depth estimation predicts a depth map from a single RGB image. (b) Classical bokeh rendering takes an image and its depth map as input to synthesize bokeh. (c) **BokehDepth** first generates a calibrated bokeh stack from the input image, and then uses the induced defocus cues to enhance depth estimation.

depth-from-defocus methods rely on multi-aperture pairs that are hard to acquire and their models often lack robustness across cameras and scenes (Ikoma et al., 2021; Favaro & Soatto, 2005; Favaro et al., 2008; Nayar et al., 2002). Modern Monocular Metric Depth Estimation (MMDE) has made rapid progress on zero-shot indoor and outdoor scenes, powered by large-scale pre-training and vision transformers (Huang et al., 2025; Bochkovskii et al., 2024; Wang et al., 2025b). Even so, these models still struggle on weakly textured distant regions and on geometrically flat surfaces (Guo et al., 2025; Gasperini et al., 2023). These are exactly the cases where defocus differences can supply an additional geometric signal that is independent of scene appearance (Ens & Lawrence, 2002; Blayvas et al., 2007; Yang et al., 2022). These limitations call for a unified, physically grounded mechanism that exploits defocus as a reliable geometric cue without tying bokeh quality to a single depth estimator.

Our core insight is to *decouple bokeh synthesis from depth prediction and leverage defocus as a supervision-free geometric cue that enhances the accuracy and physical consistency of monocular metric depth estimation*. However, turning this idea into a practical system requires us to address several coupled challenges. We must enforce physically consistent and controllable depth-free bokeh (Wadhwa et al., 2018), make defocus cues interpretable, calibratable and stable across domains (Abuolaim & Brown, 2020), integrate these signals safely with strong monocular depth foundations (Piccinelli et al., 2025; Yang et al., 2024b), and prevent noisy or spurious blur from corrupting depth predictions (Lee et al., 2021). We propose *BokehDepth*, a two stage framework shown in Figure 2 that first employs a physically grounded controllable depth-free bokeh generator to construct reliable bokeh stacks and then injects defocus cues into any monocular depth foundation model through a defocus-aware module inserted in the encoder.

In Stage-1, we build a physically guided yet depth-free bokeh generator on top of a strong pretrained image-editing backbone (Batifol et al., 2025). We unify sparse real focus–defocus pairs, in-the-wild defocused photographs with lens metadata, and synthetic bokeh renderings by mapping their defocus level to a single thin-lens-derived control scalar that measures effective bokeh strength (Fortes et al., 2025). Conditioned on this scalar, Stage-1 produces from a single sharp input a compact bokeh stack with multiple calibrated

defocus levels, without requiring any depth map. In Stage-2, we feed the Stage-1 bokeh stack and the original sharp frame into a discriminative monocular depth encoder. A lightweight bokeh-aware aggregation module is inserted into the encoder to fuse features along the calibrated bokeh-strength axis, exposing depth-sensitive defocus variations while leaving the downstream decoder and metric head unchanged. This design lets us plug **BokehDepth** into strong monocular depth foundations and turn synthetic defocus cues into consistent gains in metric accuracy and physical consistency. We summarize our contributions as follows:

- We design a bokeh renderer on top of powerful pretrained image editing backbone. Through a unified real and synthetic data pipeline and bokeh-conditioned adapters, Stage-1 generates reliable multi-strength bokeh stacks without using any depth map.
- We introduce a defocus-aware module that can be plugged into diverse monocular metric depth estimators. Given an input image and synthetic bokeh stack, it exposes stable defocus cues that enhance depth estimation.
- We show that combining Stage-1 and Stage-2 yields the *BokehDepth* framework, which improves visual fidelity over depth-map-based bokeh pipelines and consistently boosts the metric performance of strong monocular depth models across challenging indoor and outdoor scenes.

## 2. Related Works

### 2.1. Bokeh Synthesis

Defocus has been modeled with physically grounded camera and aperture formulations and with light field integration, which motivate filtering and layered reconstruction (Potmesil & Chakravarty, 1981; Kraus & Strengert, 2007; Lee et al., 2008; 2010; Vaidyanathan et al., 2015). Depth map based image space rendering uses pyramidal filtering or per pixel layered splatting but struggles near discontinuities due to occlusion and color leakage (Kraus & Strengert, 2007; Lee et al., 2008; 2010). Computational photography estimates depth from stereo or dual pixel signals and then synthesizes shallow depth of field, yet remains sensitive to segmentation and disparity errors (Barron et al., 2015; Wadhwa et al., 2018). Learning based pipelines train neural renderers for controllable bokeh, and physics guided hybrids

reduce artifacts while retaining user control (Wang et al., 2018; Xiao et al., 2018; Ignatov et al., 2020; Qian et al., 2020; Peng et al., 2022a). Differentiable and occlusion-aware renderers improve quality around edges, and layered scene representations such as multiplane images better handle partial occlusion, with extensions to video and mixed reality that enforce consistent lens characteristics (Sheng et al., 2024; Peng et al., 2022b; Mandl et al., 2024; Seizinger et al., 2025). Generative diffusion methods inject strong image priors and explicit aperture conditioning to stabilize synthesis under imperfect depth and segmentation while enabling flexible refocusing and editing (Fortes et al., 2025; Zhu et al., 2025; Wang et al., 2025c; Qin et al., 2025; Yang et al., 2025). These advances indicate that diffusion models that embed camera physics offer an artifact resistant and scalable path to scene consistent bokeh across imagery.

## 2.2. Monocular Depth Estimation

Recent advances in monocular depth estimation fall into two complementary streams, a discriminative feed-forward family and a generative diffusion family. The discriminative stream begins with end to end models that adopt a scale invariant log loss (Eigen et al., 2014) and then evolves to discretization and transformer based decoders such as DORN (Fu et al., 2018), AdaBins (Bhat et al., 2021), NeW–CRFs (Yuan et al., 2022), and iDisc (Piccinelli et al., 2023). Cross dataset transfer improves through large scale mixing in MegaDepth and MiDaS (Li & Snavely, 2018; Ranftl et al., 2020), and ZoeDepth (Bhat et al., 2023) connects relative training to metric prediction. Within this stream, camera aware modeling injects or normalizes intrinsics as in CAM–Convs (Facil et al., 2019), canonicalization with geometry branches improves absolute scale in Metric3D (Yin et al., 2023) and Metric3Dv2 (Hu et al., 2024), and high resolution detail benefits from tile based inference in PatchFusion (Li et al., 2024). Data scaling and distillation further consolidate robustness, with Depth Anything (Yang et al., 2024a) providing a broad foundation and Depth Anything V2 (Yang et al., 2024b) advancing through synthetic replacement, stronger teachers, and large pseudo labeled real images. A current focus is universal monocular metric depth that targets absolute scale without test time camera metadata. UniDepth (Piccinelli et al., 2024) and UniDepthV2 (Piccinelli et al., 2025) adopt compact designs with learned camera representations, and Depth Pro (Bochkovskii et al., 2024) estimates field of view from image features to produce sharp metric maps at high resolution. The generative stream repurposes diffusion priors for depth, where Marigold adapts Stable Diffusion for affine invariant predictions with strong zero shot transfer (Ke et al., 2024), DiffusionDepth (Duan et al., 2024) formulates depth as iterative denoising conditioned on the image, and Pixel Perfect Depth (Xu et al., 2025a) performs diffusion in pixel space with semantics prompted transformers to strengthen edges and global consistency. Our approach follows the discriminative feed-forward path for efficiency and reliability in universal metric depth while acknowledging the strengths of diffusion models in fine structure and appearance shifts.

## 3. *Stage-1*: Bokeh Generation

### 3.1. Physically Grounded Bokeh Generation

We build Stage-1 upon FLUX.1-Kontext (Batifol et al., 2025), a rectified-flow transformer that unifies text-to-image generation and instruction-guided image editing within a single latent-space backbone. This architecture provides strong priors for preserving structure and identity during complex image manipulations. To achieve lens-consistent bokeh synthesis without training a separate generator, we ground our conditioning in the thin-lens circle-of-confusion (CoC) model (Fortes et al., 2025). We map diverse optical parameters to a single calibrated scalar $K$, which approximates the linear relationship between the blur radius $r$ and the disparity offset $\Delta\text{disp}$ as $r \approx K \cdot \Delta\text{disp}$. We instantiate this control signal as

$$K(f, N, S_1) = \frac{f^2 S_1}{2N(S_1 - f)} \cdot \text{pixel\_ratio}, \qquad (1)$$

where $f$, $N$, and $S_1$ denote the focal length, aperture number, and focus distance, respectively. The term pixel_ratio converts the physical CoC diameter into target pixel units. This formulation establishes $K$ as a unified and interpretable defocus-strength axis that aligns the optical properties of real cameras with synthetic rendering logic.

To learn this continuous control space despite the scarcity of paired focus data, we train on a hybrid corpus that projects three distinct data sources into the shared $K$ domain. First, we leverage abundant in-the-wild photographs exhibiting authentic optical defocus. For these samples, we preserve EXIF metadata and derive target $K$ values by estimating the focus distance using off-the-shelf estimators. Second, we augment sharp images with physically motivated renderings from BokehMe (Peng et al., 2022a) where the control parameters are explicitly known. Third, we integrate limited paired datasets, such as DPDD (Abuolaim & Brown, 2020) and BLB (Peng et al., 2022a), by re-parameterizing their variable aperture settings into our unified scalar. This hybrid approach enables the model to learn photorealistic blur patterns from real data while retaining the precise controllability of synthetic rendering. During training, the model conditions exclusively on $K$ and alternates between text-conditional and image-conditional objectives. At inference, Stage-1 generates calibrated bokeh stacks from a single sharp image and a desired blur strength, effectively bypassing the need for explicit depth map prediction.

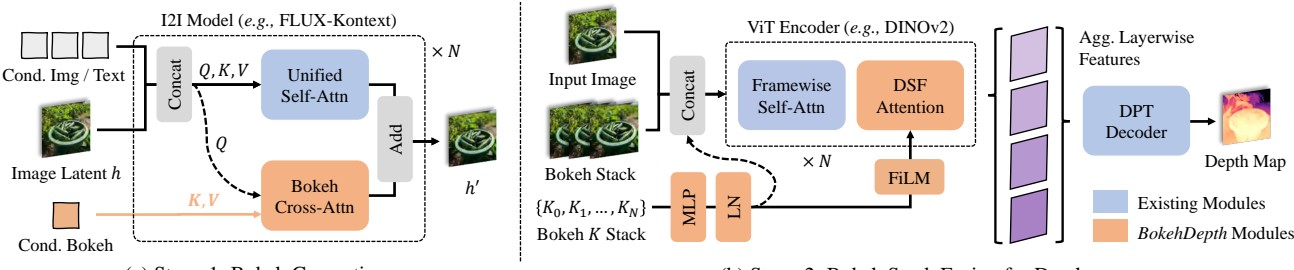

(a) Stage-1: Bokeh Generation

(b) Stage-2: Bokeh Stack Fusion for Depth

*Figure 3.* **BokehDepth** architecture. (a) Stage-1 bokeh generation augments a pretrained I2I model, such as FLUX.1-Kontext, with a bokeh cross-attention adapter that takes a scalar bokeh strength K and produces a calibrated multi-strength bokeh stack from a single sharp image. (b) Stage-2 bokeh stack fusion inserts Divided Space Focus (DSF) Attention into a ViT encoder and uses FiLM conditioning to inject the bokeh stack along the defocus axis, then feeds the aggregated layerwise features to an unchanged DPT decoder to predict metric depth.

## 3.2. Bokeh Conditioning in MMDiT Attention

Our I2I pipeline is built on a Multimodal Diffusion Transformer (MMDiT) (Esser et al., 2024), in which both text tokens and latent image tokens are processed by a single unified self-attention block rather than separate self- and cross-attention modules as in the traditional U-Net architecture (Liu et al., 2024; Hua et al., 2025; Rombach et al., 2022). Concretely, let $Q_T, K_T, V_T$ be the query, key, and value projections of the text tokens, and let $Q_I, K_I, V_I$ be the projections of the current noisy latent image tokens after timestep dependent modulation. We first concatenate the text and image branches along the token dimension, apply rotary position embedding (RoPE) (Su et al., 2024) to all queries and keys, and run one scaled dot-product attention over the joint sequence. We denote this unified attention mechanism as

$$h_{\mathrm{MMDiT}} = \mathrm{Attn}\big(\mathrm{R}(Q_T\|Q_I), \mathrm{R}(K_T\|K_I), V_T\|V_I\big), \quad (2)$$

where $\|$ denotes concatenation along the token dimension, and $\mathrm{R}(\cdot)$ denotes the application of RoPE. Inspired by Fortes et al. (2025), we control defocus by introducing a dedicated bokeh branch. A single scalar bokeh strength $K$, which specifies the desired blur magnitude per unit-disparity, is passed through a small multilayer perceptron to produce a compact conditioning vector $c_b$. Two lightweight linear projections, implemented as LoRA adapters in practice, map $c_b$ to a set of keys $K_b$ and values $V_b$ (Fortes et al., 2025). We reuse the same query that drives the unified attention in (2) and obtain a defocus-conditioned response

$$h_{\mathrm{bokeh}} = \mathrm{Attn}\big(\mathrm{R}(Q_T\|Q_I), K_b, V_b\big). \quad (3)$$

The final hidden representation after each instrumented attention block is the sum of the original multimodal interaction and the bokeh response

$$h_{\mathrm{final}} = h_{\mathrm{MMDiT}} + \lambda\, h_{\mathrm{bokeh}}. \quad (4)$$

This design enables the model to preserve the global scene layout and semantics from the image through $h_{\mathrm{MMDiT}}$, while

injecting a precise defocus control signal via $h_{\mathrm{bokeh}}$. Remarkably, this elegant bokeh-attention mechanism works without any external depth map input, yet delivers effective bokeh rendering. The architecture follows the adapter-style conditioning paradigm introduced by IP-Adapter (Ye et al., 2023), but here it is integrated into the unified attention backbone of the MMDiT-based model in the I2I editing setting.

# 4. *Stage-2*: Bokeh Stack Fusion for Depth

## 4.1. Background

### 4.1.1. DEPTH FROM DEFOCUS

Bokeh images intrinsically encode depth cues. Depth from defocus (DfD) is a long standing technique that recovers depth directly from defocus blur under a fixed viewpoint (Pentland, 1987; Suwajanakorn et al., 2015; Tang et al., 2017; Hazirbas et al., 2018; Maximov et al., 2020; Si et al., 2023; Fujimura et al., 2024; Wijayasingha et al., 2024; Xu et al., 2025b). Stage-1 of our pipeline synthesizes a bokeh stack with different blur strengths $K$ while keeping the scene, camera pose, and focus distance fixed. In theory, this stack alone is sufficient to reconstruct metric depth. We formalize this claim in the following proposition.

**Proposition 4.1** (Depth-from-Bokeh Sweep under Calibrated Bokeh Control). *For a static scene observed by a thin-lens camera with fixed pose and focus distance, we record a bokeh stack by sweeping only the calibrated bokeh strength $K$. At every pixel, the measured bokeh radius is exactly proportional to that pixel's inverse-depth offset from the focal plane. The slope of this proportionality, obtained by regressing radius on $K$ across the stack, is an unbiased and consistent estimate of that offset and yields the pixel's metric depth up to the usual front/behind-focus sign.*

A complete mathematical proof of Proposition 4.1 is given in supplementary material. The core advance over classical DfD is that Proposition 4.1 turns the defocus-to-depth relation into a per-pixel linear model and shows that the ordinary least squares slope of bokeh intensity $K$ is an unbi-

ased and statistically consistent estimator of the true inverse depth offset, which directly recovers metric depth at that pixel. In contrast, classical DfD typically takes two frames with different focus settings, estimates a blur radius, and then solves a fragile global optimization problem that needs strong priors and is sensitive to noise, weak texture, and calibration error (Schechner & Kiryati, 2000; Rajagopalan & Chaudhuri, 2002; Jin & Favaro, 2002; Ziou & Deschenes, 2001; Zhou et al., 2009; Persch et al., 2014).

### 4.1.2. DISCRIMINATIVE MONOCULAR METRIC DEPTH ESTIMATION

Modern discriminative monocular metric depth estimation follows a feed-forward design in which a single transformer pass predicts dense metric depth from one RGB view, without multi-view optimization or iterative refinement (Yang et al., 2024b; Bochkovskii et al., 2024; Piccinelli et al., 2025; Wang et al., 2025a). These systems use a large Vision Transformer encoder in the style of DINOv2 visual pretraining (Oquab et al., 2023; Darcet et al., 2023; Jose et al., 2024) together with a DPT-style decoder for dense prediction (Ranftl et al., 2021). Let $I \in \mathbb{R}^{H \times W \times 3}$ be the input RGB image. The encoder $E_\theta$ produces a multi scale feature pyramid

$$\{F^{(s)}\}_{s=1}^S = E_\theta(I), \tag{5}$$

where each $F^{(s)}$ retains global semantic context and local detail through self-attention over the entire image. A DPT-style decoder $D_\phi$ then fuses and upsamples these features to recover a full resolution depth related field

$$\hat{z} = D_\phi\left(\{F^{(s)}\}_{s=1}^S\right), \tag{6}$$

where $\hat{z}(p)$ denotes the predicted depth quantity at pixel $p$.

To express absolute metric scale, current feed-forward models attach a lightweight camera-aware head $\Gamma_\psi$ that predicts viewing geometry from the same shared features. This head estimates camera parameters such as focal-length, per-ray direction, or full intrinsics and extrinsics. We write

$$\hat{D}_{\text{metric}}(p) = \text{Scale}\left(\hat{z}(p), \Gamma_\psi(\{F^{(s)}\}_{s=1}^S)(p)\right). \tag{7}$$

where $\hat{\kappa}$ is the inferred camera representation and $\text{Scale}(\cdot)$ converts $\hat{z}$ into metric depth $\hat{D}_{\text{metric}}$ in physical units. The key idea is that the network learns both scene and viewing geometry rather than applying scale afterward.

Training across this family of models emphasizes two goals. First, globally consistent metric scale across domains through camera-aware supervision and geometric consistency constraints; Second, sharp object boundaries and fine spatial detail through edge aware and multi-scale gradient losses, often distilled from high quality synthetic depth. As

a result, these feed-forward estimators produce high resolution depth with crisp edges and reliable global scale from a single RGB frame in real time.

### 4.2. Divided Space Focus Attention in the Encoder

Our objective is to inject physically calibrated defocus cues into the discriminative monocular depth encoder while leaving the downstream decoder in Equation (7) unchanged. We assume a sharp reference RGB frame $I_0 \in \mathbb{R}^{H \times W \times 3}$ with bokeh strength $K_0 = 0$ and a synthetic bokeh stack $\{I_n\}_{n=1}^N$ produced by Stage-1 from the same viewpoint. Each $I_n$ differs only in a controllable bokeh strength $K_n \in \mathbb{R}$. Following Equation (5), a shared vision transformer encoder $E_\theta$ processes every frame independently and returns per-frame patch tokens. We denote the tokens for frame $n$ by $P_n \in \mathbb{R}^{K \times D}$ and collect them as $X = \{P_f\}_{f=0}^N \in \mathbb{R}^{(N+1) \times K \times D}$ together with the known strengths $\mathbf{K} = [K_0, \dots, K_N] \in \mathbb{R}^{(N+1) \times 1}$. Divided Space Focus Attention (DSFA) is inserted into encoder layers as an in-place feature rewriting block with **two steps**. **Step-1** performs spatial attention inside each frame. **Step-2** performs focus attention across frames at aligned spatial locations. After DSFA we retain only the refined tokens of the reference frame. The rest of the depth head can run exactly as in a standard single frame estimator.

**Step-1: spatial attention within each frame.** For each frame index $f \in \{0, \dots, N\}$ we embed the bokeh scalar $K_f$ with a learnable multilayer perceptron (MLP) $g(\cdot)$:

$$t_f = g(K_f) \in \mathbb{R}^D. \tag{8}$$

We prepend $t_f$ to that frame's $K$ patch tokens to obtain $S_f = [t_f; X_f] \in \mathbb{R}^{(1+K) \times D}$. We forward $S_f$ through a transformer block and obtain

$$\tilde{S}_f = \text{FFN}\big(\text{MSA}(\text{LN}(S_f))\big) \in \mathbb{R}^{(1+K) \times D}. \tag{9}$$

Dropping the first token yields refined patch features

$$\tilde{X}_f = \tilde{S}_f[1:] \in \mathbb{R}^{K \times D} \tag{10}$$

that encode how defocused the frame is.

**Step-2: focus attention across frames.** After spatial attention, we align patches across the stack. For each patch index $j$, we gather the same spatial location from all frames:

$$Y_j = [\tilde{X}_0[j], \tilde{X}_1[j], \dots, \tilde{X}_N[j]] \in \mathbb{R}^{(N+1) \times D}. \tag{11}$$

We modulate every element of $Y_j$ with FiLM-style conditioning (Perez et al., 2018; Dumoulin et al., 2018; Strub et al., 2018) from the same control tokens $t_f$. A learned linear map $h(\cdot)$ predicts a per-frame scale and shift

$$[a_f, b_f] = h(t_f), \quad a_f, b_f \in \mathbb{R}^D, \tag{12}$$

and we apply channel wise affine modulation

$$\hat{Y}_{f,j} = (1 + \tanh(a_f)) \odot Y_{f,j} + b_f. \qquad (13)$$

The modulated sequence $\hat{Y}_j \in \mathbb{R}^{(N+1) \times D}$ is then processed by multi-head self-attention along the frame axis. This lets each spatial location directly compare how blur changes as $K$ varies, which is the physical depth-from-defocus cue. Accordingly, every frame receives refined tokens $\bar{X}_f \in \mathbb{R}^{K \times D}$. We keep the reference representation $Z = \bar{X}_0 \in \mathbb{R}^{K \times D}$. Finally, the dense prediction head from Equation (7) upsamples $Z$ back to the pixel grid and outputs the metric depth map for the reference frame $I_0$. In effect, DSFA injects the calibrated bokeh stack into the encoder while preserving the external interface of a standard monocular metric depth estimator. Together, Stages 1 and 2 constitute the overall ***BokehDepth*** pipeline, illustrated in Figure 3.

# 5. Experiments

## 5.1. Implementation Details

**Stage-1.** Following the unified training in Section 3.1, we train our FLUX.1-Kontext-based bokeh generator for 40 epochs at a fixed resolution of $512 \times 512$, then run an additional 10 epochs using only I2I data at each dataset's native resolution to adapt to heterogeneous image sizes. Training takes 7 days on $4 \times$A6000 GPUs. We compare against classical and neural baselines BokehMe (Peng et al., 2022a), DrBokeh (Sheng et al., 2024), BokehDiff (Zhu et al., 2025), DiffCamera (Wang et al., 2025c), GenFocus (Mu et al., 2025) and the FLUX.1-Kontext editing backbone (Batifol et al., 2025) on the exposure-aligned EBB! Val200 split (Peng et al., 2023; Zhu et al., 2025; Ignatov et al., 2020). To isolate the influence of depth prediction errors, we construct a synthetic SYNTHEBOKEH300 benchmark following prior protocols (Peng et al., 2022a; Zhu et al., 2025; Sheng et al., 2024), where foreground and background layers from Lin et al. (2021) are rendered with accurate depth maps. We report PSNR and SSIM for pixel-level and structural fidelity, and LPIPS (Zhang et al., 2018) and DISTS (Ding et al., 2020) for perceptual similarity.

**Stage-2.** For metric depth estimation, we integrate the DSFA module into Depth Anything V2-L (DAv2) (Yang et al., 2024b) and UniDepthV2-L (UDv2) (Piccinelli et al., 2025). We initialize from the official L-version checkpoints and fine-tune using each base model's published training pipeline on bokeh stacks generated by our Stage-1. For cross-domain zero-shot MMDE, we fine-tune only on Hypersim (Roberts et al., 2021), where each sharp frame is paired with its synthetic bokeh stack, and evaluate alongside strong baselines on diverse indoor and outdoor benchmarks following standard $\delta_1$ and AbsRel protocols (Eigen et al., 2014; Piccinelli et al., 2025; Bochkovskii et al., 2024; Hu et al., 2024; Pham et al., 2025; Li et al., 2025; Obukhov et al.,

2025). For in-domain evaluation, we use the NYUv2 (Silberman et al., 2012) training split augmented with Stage-1 stacks and report results on the official test split.

## 5.2. Experimental Results

### 5.2.1. STAGE-1: BOKEH RENDERING

For real photographs, the quantitative results in Table 1 show that our method delivers the highest overall fidelity, with consistent gains in PSNR (Gonzalez, 2009) and SSIM (Wang et al., 2004), and markedly lower LPIPS (Zhang et al., 2018) and DISTS (Ding et al., 2020). Notably, although some of our training examples were synthesized with BokehMe (Peng et al., 2022a), the quality of our bokeh generation is not bounded by that renderer. We leverage FLUX.1-Kontext's strong image-editing priors and the real bokeh datasets to produce synthesized defocus that faithfully matches lens-like optical behavior observed in real photography.

For synthetic scenes with reliable ground-truth depth, Dr-Bokeh, which operates with direct access to depth and performs layered rendering, achieves the strongest PSNR. Our Stage-1 generator closely approaches these physically supervised baselines and attains the best perceptual quality across SSIM, LPIPS and DISTS. These results indicate that the unified, lens-derived control scalar $K$ guides our model to reproduce the spatial structure and blur profiles prescribed by geometric optics, without requiring explicit depth maps at inference time.

Taken together, the real and synthetic evaluations in Table 1 and Figure 4 show that Stage-1 learns a robust and interpretable bokeh control space. The model retains the instruction-following flexibility of FLUX.1-Kontext while lifting it to a depth-map-free yet lens-consistent bokeh renderer that generalizes across diverse natural photographs and controlled layered scenes.

### 5.2.2. STAGE-2: MONOCULAR METRIC DEPTH ESTIMATION

We now examine how DSFA strengthens existing monocular metric depth models when they are guided by Stage-1 bokeh stacks. Each depth backbone processes the sharp input image alongside a three-frame stack generated by Stage-1, featuring varying bokeh strengths $K \in [10, 30]$. Our DSFA module then aggregates these defocus cues to refine depth estimation without requiring any modifications to the original decoder or loss functions.

For cross-domain zero-shot MMDE in Table 2, the **BokehDepth** variants consistently achieve better or comparable AbsRel and $\delta_1$ than strong baselines, while broadly improving each underlying model. These gains arise because DSFA operates along a physically normalised blur axis $K$

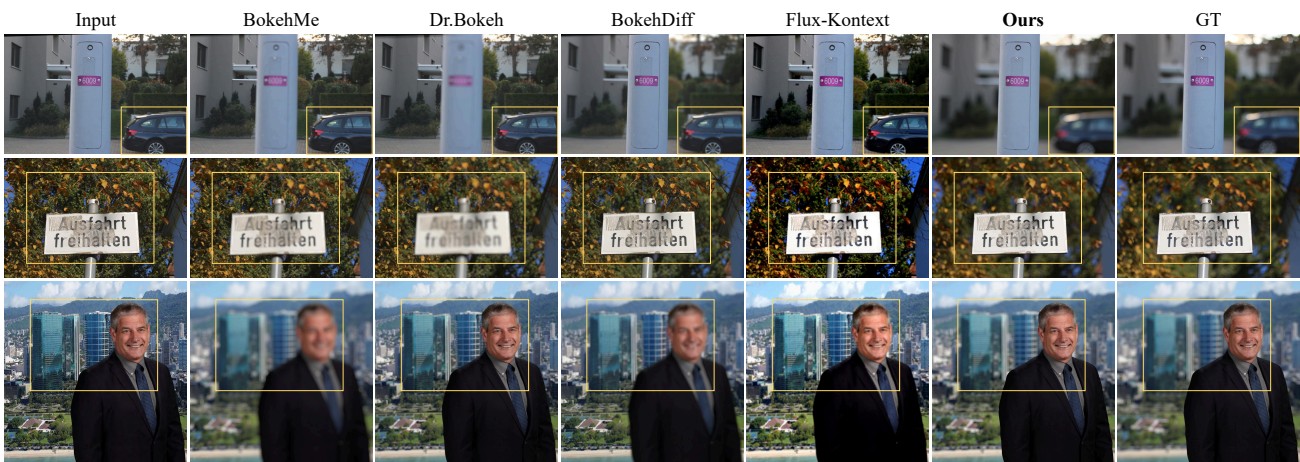

*Figure 4.* Qualitative comparisons between our Stage-1 model, BokehMe (Peng et al., 2022a), Dr. Bokeh (Sheng et al., 2024), BokehDiff (Zhu et al., 2025), FLUX.1-Kontext (Batifol et al., 2025), and the ground truth. Our method more reliably preserves in-focus subjects while producing background blur that increases monotonically with depth. At depth discontinuities, it substantially reduces edge halos and color bleeding.

| Method | EBB! Val200 (real) | | | | SYNTHEBOKEH300 (synthetic) | | | |
|---|---|---|---|---|---|---|---|---|
| | PSNR↑ | SSIM↑ | LPIPS↓ | DISTS↓ | PSNR↑ | SSIM↑ | LPIPS↓ | DISTS↓ |
| BokehMe (Peng et al., 2022a) | 24.13 | 0.751 | 0.390 | 0.143 | 22.83 | 0.809 | 0.279 | 0.188 |
| DrBokeh (Sheng et al., 2024) | 22.61 | 0.735 | 0.435 | 0.178 | **29.70** | 0.895 | 0.149 | 0.087 |
| BokehDiff (Zhu et al., 2025) | 24.35 | 0.802 | 0.279 | 0.112 | 26.21 | 0.807 | 0.288 | 0.142 |
| FLUX.1-Kontext (Batifol et al., 2025) | 19.92 | 0.645 | **0.165** | 0.427 | 20.34 | 0.685 | 0.351 | 0.155 |
| DiffCamera (Wang et al., 2025c) | 25.07 | 0.867 | 0.311 | 0.102 | 25.26 | 0.880 | 0.238 | 0.120 |
| GenFocus (Mu et al., 2025) | 25.48 | **0.886** | 0.278 | 0.081 | 21.54 | 0.895 | 0.317 | 0.153 |
| **Ours** | **25.91** | 0.856 | 0.185 | **0.076** | 29.12 | **0.901** | **0.139** | **0.073** |

*Table 1.* Quantitative comparison on EBB! Val200 (Peng et al., 2023; Ignatov et al., 2020) and SYNTHEBOKEH300. **Boldface** denotes the best and underline the second best.

rather than dataset-specific heuristics, which stabilises defocus cues across diverse scenes. On the in-domain NYUv2 benchmark, Table 3 shows that **BokehDepth** attains state-of-the-art error levels among monocular metric depth estimators. The gains are most pronounced on thin structures, reflective surfaces, and cluttered indoor layouts where single-frame cues are ambiguous: DSFA leverages structured bokeh variations to sharpen object boundaries and stabilise local metric scale while preserving the backbone's behaviour in well-conditioned regions. Compared with dedicated Depth-from-Defocus methods (Table 4), **BokehDepth** achieves the highest $\delta_1$ and the lowest RMS by a large margin—despite a strictly harder setting. Every competing DfD method constructs its evaluation focal stack by rendering RGB-D pairs through a thin-lens or PSF forward model, implicitly encoding ground-truth depth, whereas **BokehDepth** generates its bokeh stack from a single sharp image with no access to any depth map. In real-world deployment where ground-truth depth is unavailable, this advantage would widen further.

Qualitative comparisons in Figure 5 further confirm that **BokehDepth** produces depth maps with clearer layer sepa-

ration, cleaner occlusion boundaries and more reliable far-range estimates than single-frame counterparts, especially in cases where textures are weak but defocus changes remain informative. These results indicate that calibrated Stage-1 bokeh stacks together with the plug-and-play DSFA integration provide an effective bridge between learned defocus and universal MMDE.

### 5.3. Ablation Studies

**Ablation on Stage-1 Bokeh Cross-Attention Design.** By comparing FLUX.1-Kontext (Batifol et al., 2025) and our **BokehDepth** results in Table 1, we conclude that the observed improvement in bokeh rendering originates from our Stage-1 Bokeh Cross-Attention design rather than from FLUX.1-Kontext's text-driven image editing capabilities (Fortes et al., 2025; Yuan et al., 2025). The FLUX.1-Kontext base model performs poorly on physically consistent, lens-like defocus rendering. In contrast, our cross-attention design yields more accurate and visually coherent bokeh that better matches real optical defocus.

**Ablation on Stage-2 Divided Space Focus Attention.** Ta-

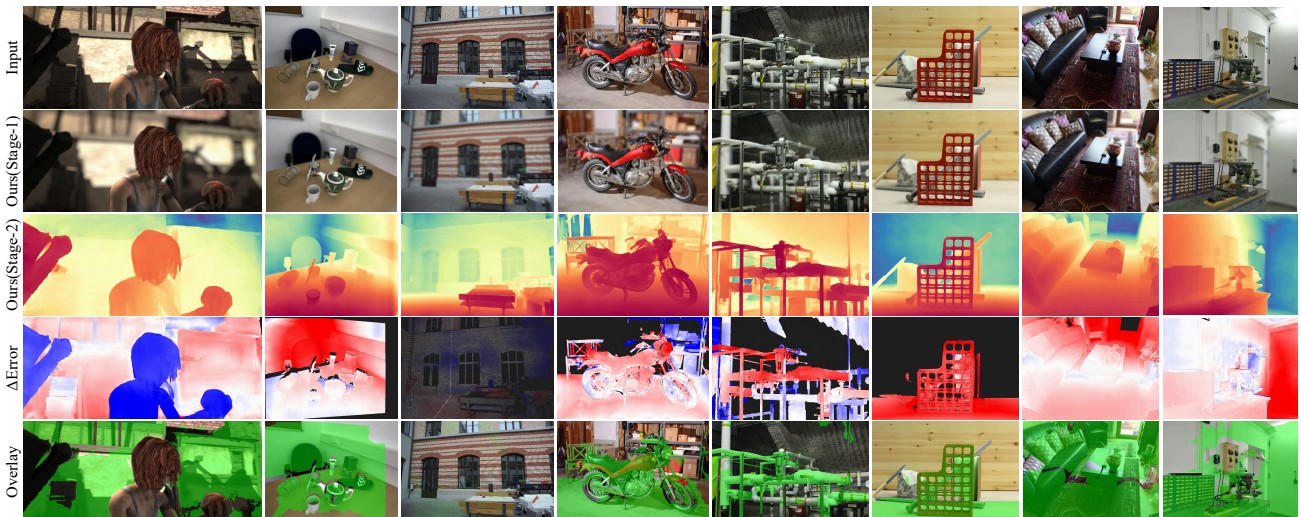

*Figure 5.* Qualitative results of **BokehDepth** . *Top to bottom:* input image, a frame from the Stage-1 bokeh stack, predicted depth by Stage-2, ΔError maps showing per-pixel reduction in absolute depth error of **BokehDepth** relative to base model Depth Anything V2 (Yang et al., 2024b), and RGB images overlaid with green highlights indicating improvement regions. **BokehDepth** lowers depth errors on fine structures, weakly-textured walls and distant background regions, offering more distinct layer separation and steadier metric depth across varied scenes.

| Method | HAMMER | | IBims-1 | | Middlebury | | Make3D | | Sintel | | ETH3D | |
|--------|--------|------|---------|------|------------|------|--------|------|--------|------|-------|------|
| | $\delta_1\uparrow$ | Abs↓ | $\delta_1\uparrow$ | Abs↓ | $\delta_1\uparrow$ | Abs↓ | $\delta_1\uparrow$ | Abs↓ | $\delta_1\uparrow$ | Abs↓ | $\delta_1\uparrow$ | Abs↓ |
| ZoeDepth | 0.009 | 0.943 | 0.672 | 0.174 | 0.538 | 0.214 | 0.649 | 0.221 | 0.078 | 0.946 | 0.338 | 0.500 |
| Metric3Dv2 | 0.653 | 0.357 | 0.941 | 0.100 | 0.299 | 0.450 | 0.512 | 0.394 | 0.383 | **0.370** | **0.987** | **0.042** |
| DepthPro | 0.630 | 0.391 | 0.823 | 0.159 | 0.605 | 0.251 | 0.728 | 0.254 | 0.400 | 0.508 | 0.415 | 0.327 |
| DAv2 | 0.828 | 0.134 | 0.938 | 0.080 | 0.618 | 0.240 | 0.726 | 0.217 | 0.538 | 0.569 | 0.910 | 0.091 |
| **+ Ours** | 0.894 | 0.105 | 0.960 | 0.064 | 0.675 | 0.215 | 0.740 | **0.206** | 0.542 | 0.413 | 0.954 | 0.077 |
| UDv2 | 0.645 | 0.293 | 0.945 | 0.082 | 0.535 | 0.288 | 0.739 | 0.263 | 0.344 | 0.496 | 0.852 | 0.160 |
| **+ Ours** | **0.895** | **0.094** | **0.978** | **0.039** | **0.716** | **0.205** | **0.786** | 0.228 | **0.671** | 0.391 | 0.963 | 0.074 |

*Table 2.* Zero-shot metric-depth comparison across diverse indoor and outdoor datasets (Jung et al., 2022; Koch et al., 2018; Scharstein et al., 2014; Saxena et al., 2008; Butler et al., 2012; Schops et al., 2017). **Boldface** denotes best, underline second best. $\delta_1$: accuracy threshold; Abs: Abs_Rel error.

| Method | Abs_Rel↓ | RMS↓ | Log₁₀ ↓ |
|--------|----------|------|----------|
| ZoeDepth (Bhat et al., 2023) | 0.077 | 0.278 | 0.033 |
| Metric3Dv2 (Hu et al., 2024) | 0.047 | 0.183 | 0.020 |
| DepthAnythingv2 (Yang et al., 2024b) | 0.056 | 0.206 | 0.024 |
| UniDepthV2 (Piccinelli et al., 2025) | 0.047 | 0.180 | 0.020 |
| **Ours** | **0.039** | **0.043** | **0.016** |

*Table 3.* Metric depth comparison of **BokehDepth** (UniDepthV2 backbone) against other monocular metric depth estimation methods on the NYUv2 validation set (Silberman et al., 2012). All models were trained or fine-tuned on NYUv2.

| Method | $\delta_1\uparrow$ | Abs_Rel↓ | RMS↓ |
|--------|--------|----------|------|
| Deep-Optics (Chang & Wetzstein, 2019) | 0.930 | 0.087 | 0.433 |
| DFF-DFV (Yang et al., 2022) | 0.967 | 0.445 | 0.232 |
| 2HDED:Net (Nazir et al., 2023) | 0.914 | 0.029 | 0.244 |
| DAIF-Net (Si et al., 2023) | 0.950 | 0.170 | 0.325 |
| SDNet (Zhang et al., 2025) | 0.964 | **0.026** | 0.201 |
| **Ours** | **0.978** | 0.039 | **0.043** |

*Table 4.* Metric depth comparison of **BokehDepth** (UniDepthV2 backbone) against other Depth-from-Defocus (DfD) methods on the NYUv2 validation set (Silberman et al., 2012). All models were trained or fine-tuned on NYUv2. Apart from ours, every DfD method listed above constructs its evaluation focal stack by rendering NYUv2 RGB-D pairs through a thin-lens or PSF forward model.

ble 5 ablates the main components of the Stage-2 DSFA module. *BokehDepth (w/o Focus)* keeps only the Space branch and removes the Focus branch. *BokehDepth (w/o Space)* drops the Space branch so the Focus branch alone performs cross-frame attention. *BokehDepth (w/o FiLM)* keeps both branches but disables FiLM conditioning on the focus parameter $k$. *BokehDepth (w/o Bokeh)* repeats the all-in-focus image across the stack with $K=0$, removing meaningful defocus variation. *BokehDepth (w/ BokehMe)* feeds

DSFA with bokeh stacks rendered by BokehMe (Peng et al., 2022a) using ground-truth sparse LiDAR depth instead of Stage-1 outputs. *BokehDepth (w/ Pred-BokehMe)* replaces the ground-truth depth with the dense depth map predicted by the base DAv2 model itself, providing BokehMe with spatially complete input. *BokehDepth (w/ FLUX.1-Kontext)*

| Method | $\delta_1 \uparrow$ | Abs_Rel$\downarrow$ |
|---|---|---|
| Depth Anything v2 (Yang et al., 2024b) | 0.914 | 0.097 |
| BokehDepth (w/o Focus) | 0.620 | 0.215 |
| BokehDepth (w/o Space) | 0.567 | 0.236 |
| BokehDepth (w/o FiLM) | 0.602 | 0.223 |
| BokehDepth (w/o Bokeh) | 0.855 | 0.119 |
| BokehDepth (w/ BokehMe) | 0.914 | 0.094 |
| BokehDepth (w/ Pred-BokehMe) | 0.918 | 0.094 |
| BokehDepth (w/ FLUX.1-Kontext) | 0.608 | 0.227 |
| **Ours** | **0.943** | **0.084** |

*Table 5.* Ablation study of Stage-2 DSFA. Trained on VKITTI2 (Cabon et al., 2020), evaluated on KITTI (Geiger et al., 2012) Eigen split.

| Component | Params | Lat. (s/img) | Mem (GiB) |
|---|---|---|---|
| UniDepthV2 (baseline) | 353.83M | 0.045 | 2.48 |
| UniDepthV2 + DSFA | 464.07M | 0.114 | 3.25 |
| Stage-1 bokeh rendering | 12.32B | 48.663 | 33.52 |
| Full pipeline | 12.78B | 48.777 | 33.52 |

*Table 6.* Computational cost breakdown of BokehDepth (UniDepthV2 backbone). Stage-1 runs 30 diffusion steps and operates entirely offline. Once the bokeh stack is precomputed, DSFA adds minimal overhead to the baseline.

bypasses Stage-1 entirely and feeds DSFA with raw bokeh renderings produced by the unmodified FLUX.1-Kontext backbone without our Bokeh Cross-Attention conditioning. Two observations emerge from these results. First, the bokeh source must be spatially dense and cross-frame consistent for DSFA to extract useful defocus cues. Pred-BokehMe slightly outperforms BokehMe because dense predicted depth yields smoother blur boundaries than sparse LiDAR, yet both fall well short of Stage-1, which produces inherently dense and consistent stacks without relying on any depth prior. Second, raw generative blur is not sufficient. FLUX-Kontext without our calibration mechanism collapses to $\delta_1=0.608$, far below even the no-bokeh baseline. This confirms that DSFA does not exploit appearance-level artifacts of the generative backbone but instead requires a physically calibrated defocus sweep aligned along the $K$ axis, which is exactly what Stage-1 provides.

**Computational Cost Analysis.** Table 6 profiles the computational overhead of each component. Stage-2 DSFA introduces minimal additional parameters and latency on top of the baseline, representing a marginal cost relative to the accuracy gains reported in Tables 2–4. Stage-1 dominates the end-to-end runtime because it runs the full FLUX.1-Kontext backbone for 30 diffusion steps per bokeh stack. Crucially, Stage-1 operates as a fully offline data-generation step analogous to synthetic augmentation, and its outputs can be precomputed, cached, and reused across arbitrary downstream depth models. Once the bokeh stacks are cached, the online inference cost of BokehDepth remains nearly identical to that of the unmodified baseline. We note that the current

Stage-1 latency stems from the unoptimized multi-step diffusion schedule rather than from an inherent architectural bottleneck. Applying one-step or few-step distillation (Yin et al., 2024; Sauer et al., 2024) to the Stage-1 generator offers a direct path toward real-time bokeh-stack synthesis.

# 6. Conclusion

***BokehDepth*** asks whether lens-aware defocus, grounded in thin-lens physics and powered by a pretrained image-editing prior, can act as a depth-map-free geometric cue for monocular metric depth. We propose a two-stage framework that first learns a controllable bokeh generator on FLUX.1-Kontext to synthesize calibrated bokeh stacks without depth maps, and then feeds these stacks into existing depth encoders through a plug-and-play Divided Space Focus Attention module. Across real and synthetic bokeh benchmarks and diverse indoor and outdoor depth datasets, this design improves visual fidelity over depth-based bokeh pipelines and consistently boosts strong monocular depth foundations in both in-domain and zero-shot settings. These results suggest that physically guided defocus can serve as a scalable and supervision-free signal that complements large image backbones and moves toward unified models that jointly learn image formation, defocus and depth.

# Acknowledgement

This research is supported by the National Research Foundation, Singapore, under its NRF Fellowship Award NRF-NRFF16-2024-0003 and NTU SUG-NAP. This research is also supported by cash and in-kind funding from NTU S-Lab and industry partner(s).

# Impact Statement

This paper aims to advance monocular metric depth estimation by leveraging controllable defocus cues synthesized as a bokeh stack and fusing them into a depth prediction backbone. Improved single-camera depth perception can benefit a wide range of applications such as robotics navigation, augmented reality, accessibility assistance, and content creation, especially in settings where multi-sensor hardware is impractical. At the same time, more accurate monocular metric depth estimation may be misused for intrusive 3D reconstruction or surveillance, and model failures under challenging conditions such as motion, exposure variation, rolling shutter, or focus drift could pose risks in safety-critical deployments. We encourage responsible use that respects privacy norms and legal constraints, and we emphasize that additional validation is necessary before using the method in high-stakes real-world systems.

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

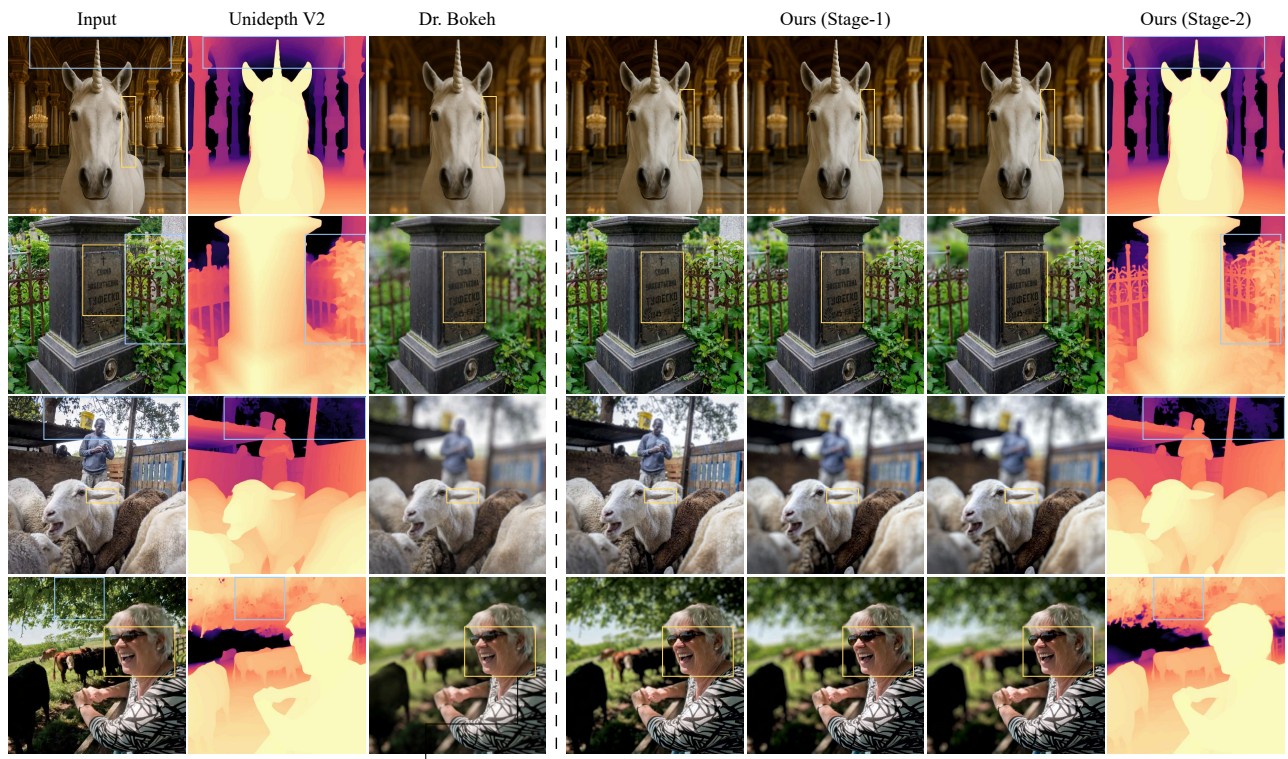

*Figure 6.* More qualitative results of **BokehDepth** . *Left:* conventional pipelines predict depth from a single sharp image and render bokeh from the noisy depth map. *Right:* our two-stage framework, where Stage-1 generates a calibrated bokeh stack with multiple bokeh strengths from a single image and Stage-2 built on UniDepthV2 (Piccinelli et al., 2025) fuses defocus cues to produce sharper and more reliable metric depth.

## A. Stage-1 Background Details

### A.1. FLUX.1-Kontext

FLUX.1-Kontext is a rectified flow transformer that unifies text-to-image (T2I) synthesis and instruction guided image editing in a single model (Batifol et al., 2025). Earlier systems such as SDXL (Rombach et al., 2022; Podell et al., 2023) or PixArt (Chen et al., 2023; 2024) typically keep generation and editing in separate networks or attach task specific adapters to a base T2I model. In contrast, FLUX.1-Kontext uses one backbone for both tasks. The model operates in the latent space of a learned autoencoder (Rombach et al., 2022; Batifol et al., 2025): an input RGB image is encoded into a spatial latent grid, and all generation and editing is performed in that latent space before decoding back to pixels (Podell et al., 2023).

Instead of classical denoising diffusion, which learns to reverse a long noise corruption process through many discrete denoising steps (Sohl-Dickstein et al., 2015; Ho et al., 2020; Rombach et al., 2022; Podell et al., 2023), FLUX.1-Kontext follows the flow-matching formulation (Liu et al., 2022; Esser et al., 2024; Batifol et al., 2025). Let $x_0 \sim q(x_0)$ be a clean latent sample from the data distribution and let $\varepsilon \sim \mathcal{N}(0, \mathbf{I})$ be Gaussian noise. We define a straight interpolation path between data and noise

$$x_t = (1 - t)\, x_0 + t\, \varepsilon, \quad t \in [0, 1], \tag{14}$$

$$\frac{dx_t}{dt} = v_\theta(x_t, t, c), \tag{15}$$

where $v_\theta$ is a time dependent velocity field predicted by a transformer under conditioning $c$. The model learns a velocity field that transports noise to data along this nearly linear path (Liu et al., 2022; Esser et al., 2024). Because $x_t$ is a convex combination of $x_0$ and $\varepsilon$, the ideal instantaneous velocity is $(\varepsilon - x_0)$, which is constant in $t$. The training objective is

$$\min_\theta\ \mathbb{E}_{t, x_0, \varepsilon, c}\, \|(\varepsilon - x_0) - v_\theta(x_t, t, c)\|_2^2 . \tag{16}$$

At inference time, sampling integrates the learned ODE from $t = 1$ (noise) to $t = 0$ (clean latent) using only a few solver steps (Liu et al., 2022; Esser et al., 2024; Batifol et al., 2025). This yields fast image generation while preserving structural sharpness, readable text, and consistent identity.

For conditional generation and editing, FLUX.1-Kontext learns the conditional distribution $p(x_0 \mid y, c)$, where $y$ is an optional visual reference such as a style or identity exemplar and $c$ is a natural language instruction (Batifol et al., 2025). The current editable canvas $x_0$, all reference images $y$, and the instruction $c$ are encoded into tokens and concatenated into a single multimodal sequence processed by a large rectified-flow transformer related to the Multimodal Diffusion Transformer (MMDiT) that was introduced for high resolution text to image synthesis under rectified flow training (Esser et al., 2024; Batifol et al., 2025). Rotary Position Embedding (RoPE) (Su et al., 2024) is generalized to three coordinates $(t, h, w)$, where $(h, w)$ are spatial indices and $t$ marks the source stream such as the editable canvas, each visual reference, or the text instruction. This positional encoding allows the transformer to attend across all sources in one pass while preserving both spatial layout and source identity (Batifol et al., 2025). A single backbone can therefore support pure generation, style transfer, identity preserving editing and iterative refinement in context across multiple user turns (Batifol et al., 2025).

Editing in FLUX.1-Kontext is formulated as conditional generation rather than masked inpainting (Batifol et al., 2025), which will be discussed in detail in Section 3.2. The model receives the editable canvas together with optional visual references and the text instruction and aligns them in latent space using scaled dot product attention (Vaswani et al., 2017). Let $Q$, $K$, and $V$ denote the query, key and value projections of the concatenated tokens. Attention is computed as

$$\text{Attention}(Q, K, V) = \text{Softmax}\left(\frac{QK^\top}{\sqrt{d}}\right) V. \tag{17}$$

During sampling, a rectified flow solver integrates the transformer's learned velocity field to update the latent canvas under the fused conditioning signals, enabling precise edits, coherent global restyling, and stable subject identity across successive user-guided refinements (Liu et al., 2022; Esser et al., 2024; Batifol et al., 2025).

## A.2. Thin Lens Bokeh & Synthetic Bokeh Rendering

When a camera images a three dimensional scene onto a two dimensional sensor, only points at one specific focus distance appear perfectly sharp. Points in front of or behind that distance form small blurred disks on the sensor. These disks are known as circles of confusion (CoC) and they give rise to the familiar out of focus bokeh pattern in photography (Lagendijk & Biemond, 2009; Wadhwa et al., 2018; Yang et al., 2016; Peng et al., 2022a; Sheng et al., 2024). Under the thin lens model, the diameter $d$ of the CoC depends on the lens geometry and the scene depth (Lagendijk & Biemond, 2009). For a lens of focal length $f$ and aperture $N$ (that is, f number $N = f/A$ with entrance pupil diameter $A$), focused at distance $S_1$ while observing a subject at distance $S_2$, the blur diameter is

$$d = \frac{f^2}{N(S_1 - f)} \frac{|S_2 - S_1|}{S_2}, \tag{18}$$

where $S_1$ is often called the focus distance and $S_2$ is the subject distance. Larger focal length $f$, wider aperture (smaller $N$), and greater separation between $S_1$ and $S_2$ all increase $d$, which produces stronger background blur and a more pronounced bokeh appearance (Lagendijk & Biemond, 2009; Yang et al., 2016; Wadhwa et al., 2018).

Classical synthetic defocus methods approximate this physical effect by spreading, or splatting, each pixel over a disk whose radius matches its local blur size (Yang et al., 2016; Peng et al., 2021). Given per pixel disparity, the blur radius $r$ can be written as

$$r = K \cdot \Delta\text{disp}, \quad \Delta\text{disp} = \left|\frac{1}{S_1} - \frac{1}{S_2}\right|, \tag{19}$$

where $K$ is a camera-dependent scale that absorbs intrinsic parameters such as focal length and aperture (Yang et al., 2016; Wadhwa et al., 2018). Pixels near the current focus distance $S_1$ receive a small blur radius and stay sharp, while pixels far from $S_1$ receive a large radius and become strongly blurred (Yang et al., 2016; Peng et al., 2021).

This splatting model produces convincing shallow depth of field in regions that are smooth in depth, but it often fails near strong occlusion boundaries. At such boundaries, foreground colors can leak into the blurred background or background colors can wash across the foreground edge, which creates unnatural halos or color bleeding artifacts (Wadhwa et al., 2018; Peng et al., 2022a). Modern bokeh renderers address this limitation by making the process explicitly aware of occlusion and

layering. One line of work uses neural refinement modules to inpaint or correct contaminated regions after classical splatting (Peng et al., 2022a). Another line decomposes the scene into ordered depth layers or multiplane images and composites them from back to front with explicit handling of partial occlusion. This layered or occlusion aware rendering strategy suppresses color bleeding at depth discontinuities and yields more realistic subject boundaries and foreground isolation (Peng et al., 2022a; Sheng et al., 2024).

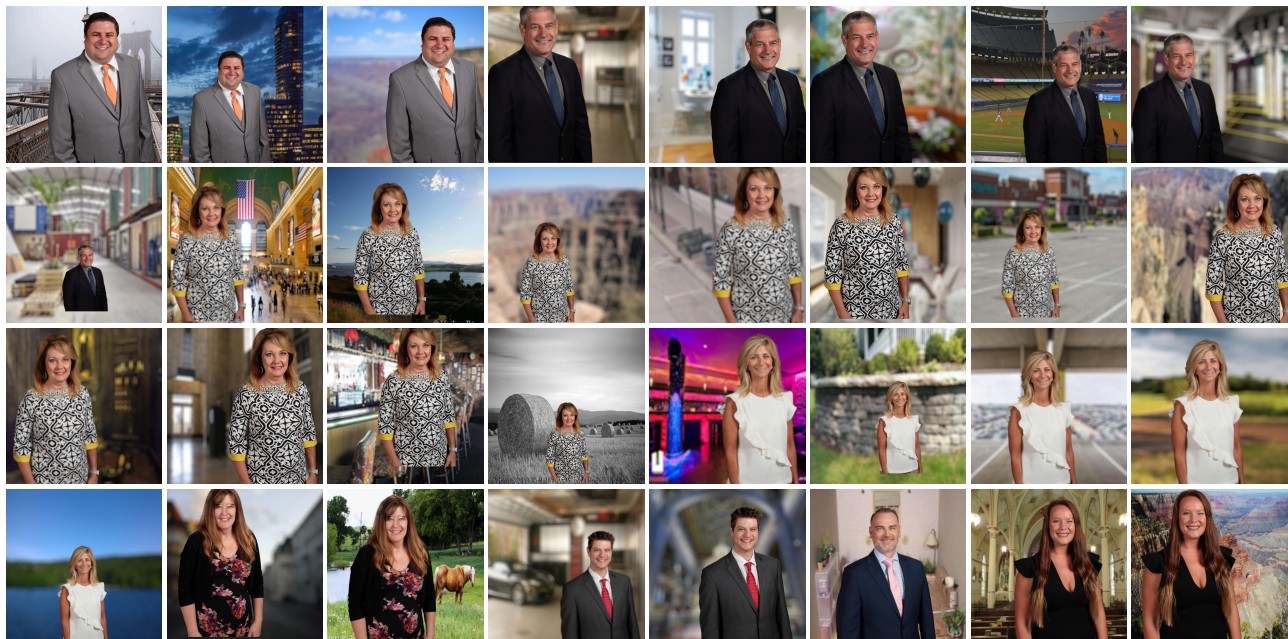

*Figure 7.* Illustration of some samples from the SYNTHEBOKEH300 validation set.

## B. Stage-1 Dataset & Unified Training Details

### B.1. T2I Dataset Details

We build on the hybrid dataset construction strategy introduced in Bokeh Diffusion (Fortes et al., 2025) and adapt it to our T2I training setting. The first branch is a curated in-the-wild subset of roughly fifteen thousand Flickr-style photographs with permissive licenses. For each image, we record EXIF metadata capturing focal length and aperture, estimate dense metric depth using DepthPro (Bochkovskii et al., 2024), extract a high-resolution foreground mask (Zheng et al., 2024), and obtain descriptive captions from large vision-language models. We remove samples with unreliable EXIF metadata, degenerate depth, or trivial foreground regions. About ten percent of the images are nearly all-in-focus and serve as sharp anchors.

The second branch is a synthetic augmentation subset. For each nearly all-in-focus anchor we render bokeh using a physically motivated defocus renderer such as BokehMe (Peng et al., 2022a). We estimate per-pixel disparity, sample a blur strength, and construct contrastive pairs that differ only in depth of field. These pairs teach the model that the same scene layout can appear either sharp or softly blurred, which is essential for controllable editing (Fortes et al., 2025).

To train with a single conditioning signal, we express both real and rendered blur using a common scalar parameter $K$. Starting from the thin-lens model, the circle-of-confusion diameter for a point at subject distance $S_2$ when focusing at $S_1$ with focal length $f$ and aperture $N$ is

$$d = \frac{f^2}{N(S_1 - f)} \frac{|S_2 - S_1|}{S_2}. \tag{20}$$

Rewriting $d$ in terms of the disparity difference $\Delta\text{disp} = \left|\frac{1}{S_1} - \frac{1}{S_2}\right|$ gives a radius proportional to $\Delta\text{disp}$,

$$K \approx \frac{f^2 S_1}{2N(S_1 - f)} \times \text{pixel\_ratio}, \tag{21}$$

where pixel_ratio accounts for sensor pitch and target resolution. We compute $S_1$ from the median depth of the salient foreground region because EXIF metadata seldom records focus distance, and we discard scenes with unstable inferred $S_1$. This conversion assigns each real photograph a physically grounded $K$ consistent with the synthetic rendering parameter used by BokehMe, allowing real and synthetic samples to be mixed within a single training batch (Fortes et al., 2025; Peng et al., 2022a).

Finally, we explain why we rely on this hybrid recipe instead of only using multi-aperture collections such as DPDD (Abuolaim & Brown, 2020). DPDD provides around five hundred controlled DSLR scenes with paired blurred and all-in-focus captures. These scenes are mostly static tabletop or indoor arrangements and rarely contain people in motion or crowded outdoor environments, which limits coverage. Our in-the-wild branch covers dynamic subjects, complex human-centric shots, and diverse compositions, while the synthetic branch supplies precise focus and defocus supervision with continuous aperture control. Together they yield a dataset that supports physically grounded bokeh control and identity-preserving editing in our model (Fortes et al., 2025).

### B.2. Scene-consistent T2I Bokeh Generation

To ensure that the T2I synthesized bokeh stack maintains consistent scene structure and subject identity across varying defocus levels, we adopt the generation pipeline proposed in Bokeh Diffusion (Fortes et al., 2025). This framework employs a grounded self-attention mechanism to anchor the spatial layout to a pivot image and utilizes a color transfer step to harmonize illumination statistics. Although our inference stage targets image-to-image editing, we implement part of the training process as a text-to-image task. This strategy allows us to leverage extensive unpaired in-the-wild photography to learn robust optical properties rather than relying solely on limited paired supervision.

### B.3. Bokeh Strength Alignment Across I2I datasets

Stage-1 I2I training uses a single scalar bokeh strength $K$ to control the blur magnitude per unit-disparity, matching the thin-lens formulation in Equation (21). To mix real photographs, captured triplets, and synthetic renders in a single model, we convert all image-to-image bokeh datasets into a unified JSONL schema. Each row stores the input all-in-focus image, the target bokeh image, an optional depth map, and a camera_anns dictionary that contains the aperture $N$, focal length $f_{\mathrm{mm}}$, 35 mm-equivalent focal length $f_{35\mathrm{mm}}$, sensor width, focus distance $S_1$, and a calibrated bokeh strength field dof-cond. A crop-corrected variant dof-cond-crop records the same quantity normalized to a full-frame field of view. All $K$ values are obtained by calling a single thin-lens utility calc_dof_cond, which implements Equation (21) and returns the expected circle-of-confusion diameter in pixels for a given configuration, then normalizes it to a reference width of 512.

**DPDD indoor Canon CR2.** The DPDD indoor split contains Canon RAW image pairs captured with different $f$-numbers under fixed scene geometry. Our conversion script calls exiftool on all .CR2 files and parses CreateDate, FNumber, FocalLength, ApproximateFocusDistance, FocusDistanceUpper, FocusDistanceLower, and the focal-plane resolution and image dimension fields. For each exposure, we estimate the focus distance $S_1$ in meters by using ApproximateFocusDistance when present and otherwise averaging FocusDistanceUpper and FocusDistanceLower. We sort images by time, form candidate pairs within a short temporal window that match in focal length and focus distance, and designate within each group the smallest $f$-number as the bokeh target and the largest $f$-number as the all-in-focus source.

For every selected bokeh shot we estimate the sensor width in millimeters in a multi-step fashion. When focal-plane resolution and resolution units are available, we convert pixels to millimeters using the reported pitch. If this fails, we fall back to inverting $f_{\mathrm{mm}}$ and $f_{35\mathrm{mm}}$ through calc_sensor_width, and finally use camera-model heuristics for common Canon full-frame and APS-C bodies based on the EXIF Model string and image resolution. With $N$, $f_{\mathrm{mm}}$, $f_{35\mathrm{mm}}$, $S_1$, and the target image size $(W, H)$ in hand, we call calc_dof_cond to obtain a physically grounded blur slope $K$ at the native resolution. This value is rescaled to a reference width of 512 by multiplying with $512/W$ and stored in dof-cond. When both $f_{\mathrm{mm}}$ and $f_{35\mathrm{mm}}$ are known, we also compute the crop factor and divide dof-cond by this factor to obtain a full-frame aligned value dof-cond-crop. Finally, a relaxed depth-of-field test based on the near and far bounds in Equation (21) and a sensor-size-dependent circle-of-confusion estimate marks each row with a boolean foreground_clear flag, which indicates whether a near-foreground plane remains inside the in-focus region.

**Aperture-Dataset.** The Aperture-Dataset contains captured triplets with a fixed scene and focus setting and three apertures: a small aperture ($f/22$) that is close to pinhole imaging and two bokeh views ($f/8$ and $f/2$) that share a common metric

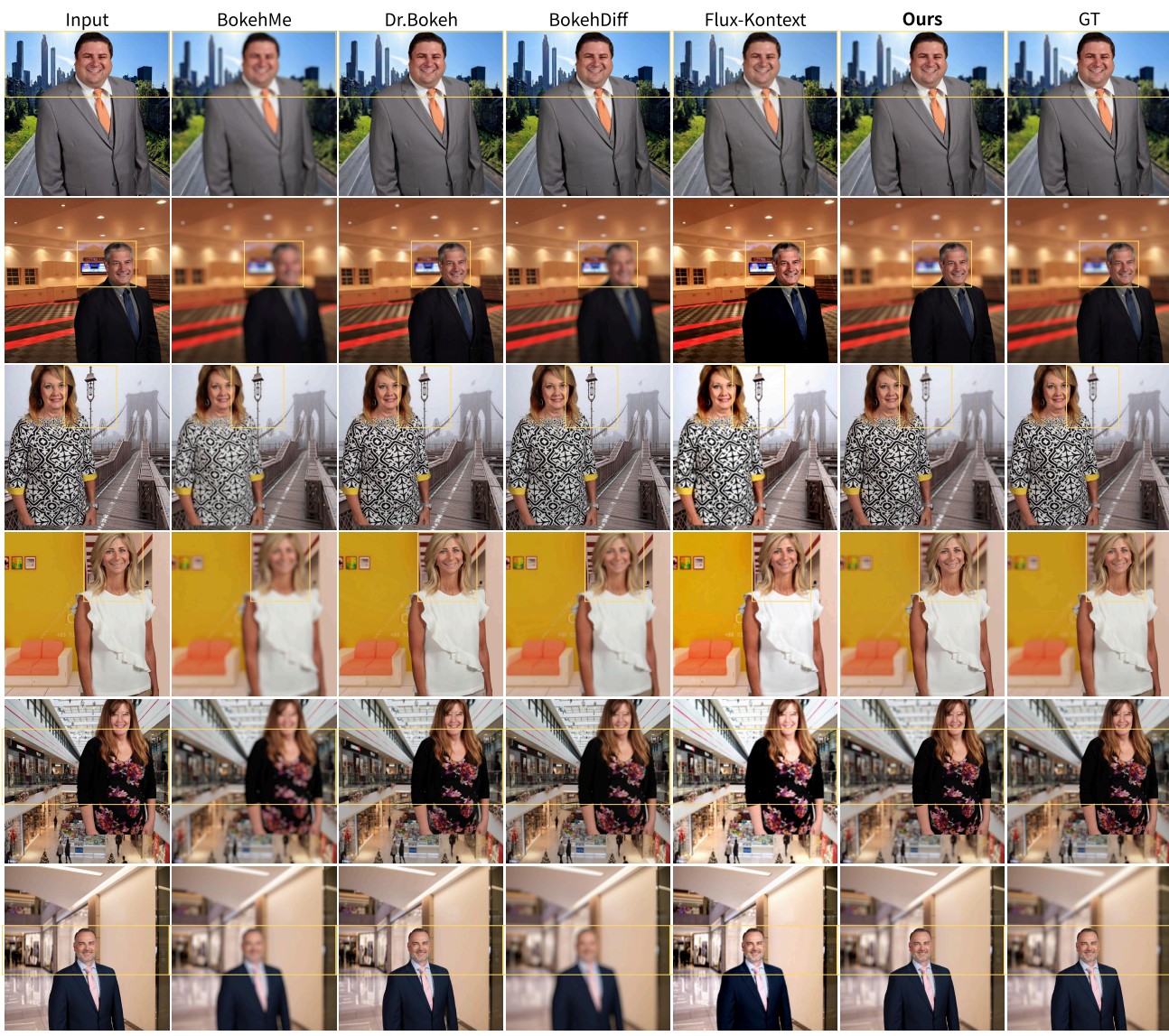

*Figure 8.* Qualitative comparisons for SYNTHEBOKEH300 (synthetic) between our Stage-1 model, BokehMe (Peng et al., 2022a), Dr. Bokeh (Sheng et al., 2024), BokehDiff (Zhu et al., 2025), FLUX.1-Kontext (Batifol et al., 2025), and the ground truth. Our method more reliably preserves in-focus subjects while producing background blur that increases monotonically with depth. At depth discontinuities, it substantially reduces edge halos and color bleeding. Overall, the rendered bokeh is visually closer to the ground truth.

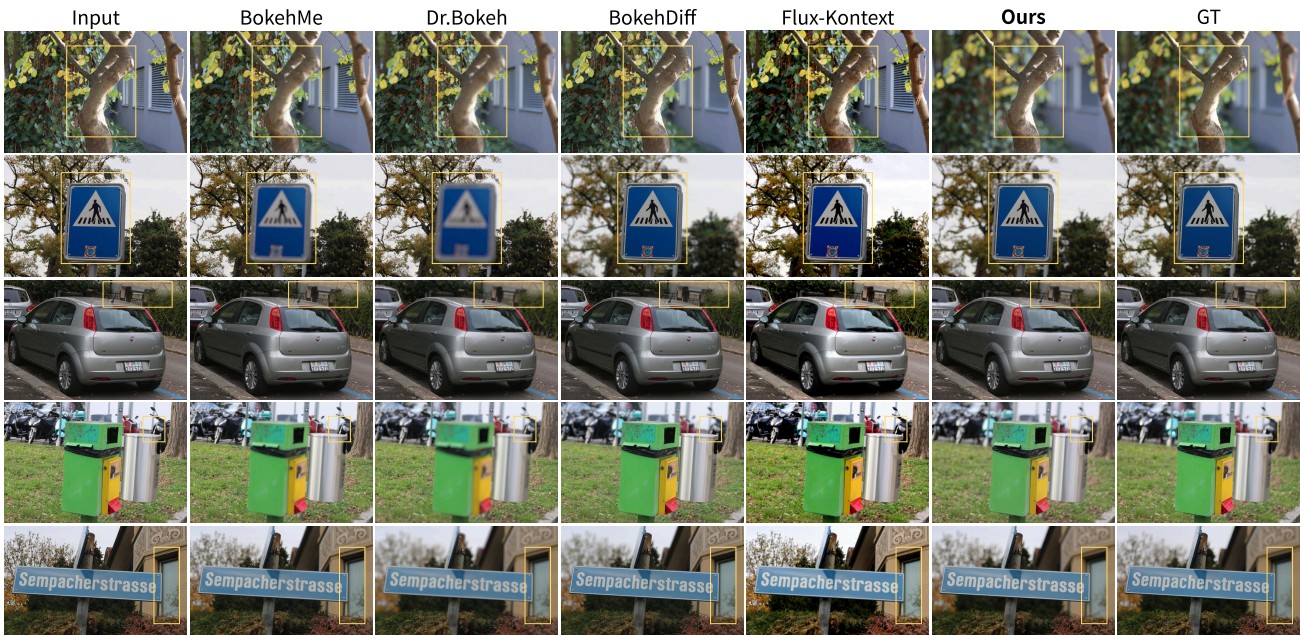

*Figure 9.* Qualitative comparisons for EBB! Val200 (real) (Peng et al., 2023; Ignatov et al., 2020) between our Stage-1 model, BokehMe (Peng et al., 2022a), Dr. Bokeh (Sheng et al., 2024), BokehDiff (Zhu et al., 2025), FLUX.1-Kontext (Batifol et al., 2025), and the ground truth. Our method more reliably preserves in-focus subjects while producing background blur that increases monotonically with depth. At depth discontinuities, it substantially reduces edge halos and color bleeding. Overall, the rendered bokeh is visually closer to the ground truth.

depth map. We convert this dataset by treating the $f/22$ image as the input all-in-focus source and the $f/8$ or $f/2$ image as the target bokeh output. Since focus distance is not stored in EXIF, we estimate $S_1$ from the depth map. For each scene, we resize the depth map to match the $f/22$ view, compute image gradients on the sharp RGB image, and form a gradient-weighted median over depth values along high-frequency edges. This yields a robust estimate of the physical focus distance that concentrates on visually salient structures.

Camera parameters for Aperture-Dataset are set through explicit defaults. We assume a Canon EOS body with a $50\,\mathrm{mm}$ lens, full-frame sensor width of $36\,\mathrm{mm}$, and crop factor $1.0$, while allowing these values to be overridden by command-line arguments. Given $N_{\mathrm{bokeh}} \in \{8.0, 2.0\}$, $f_{\mathrm{mm}}$, $f_{35\mathrm{mm}}$, the recovered $S_1$, and the target image size, we compute $K$ with `calc_dof_cond` and again normalize it to a width of $512$. The resulting values are stored as `dof-cond` and `dof-cond-crop` in `camera_anns` for each bokeh level. The same relaxed depth-of-field criterion as in DPDD is applied to label `foreground_clear`. Although both bokeh targets share the same all-in-focus source and focus distance, their different $f$-numbers produce distinct $K$ values, which lets the model learn how bokeh strength changes as the physical aperture opens.

**BLB synthetic renders.** The BLB dataset provides fully rendered defocus stacks with known intrinsics and focus distances. Each scene directory contains a sharp RGB image, a disparity map, a set of bokeh renders indexed by focus and aperture, and an `info.json` file listing the physical parameters: focus distances, $f$-stop values, focal length, sensor width, and rendered resolution. Our pipeline converts `disparity.jpg` into an approximate metric depth map (`.npz`), treats `focal_length` and `sensor_width` as ground truth, converts them to millimeters, and computes a 35 mm-equivalent focal length and crop factor.

The renderer supplies a normalized $f$-stop rather than a conventional $f$-number. We map each $f$-stop to a continuous $f$-number between roughly $f/1.4$ and $f/16$ using logarithmic interpolation over a standard aperture series, and use this value as $N$ in Equation (21). For each valid focus–aperture pair with an available bokeh render, we compute $K$ at two resolutions using the same thin-lens utility. We call `calc_dof_cond` at the native BLB resolution ($W_{\mathrm{orig}}, H_{\mathrm{orig}}$) for the FLUX branch. Outputs are clipped to $[0, 30]$ for stability and stored as `dof-cond` (native) and `dof-cond-crop` (512-normalized). We also compute a scalar `disp_focus` capturing the scene's relative focus depth and combine it with an analytic depth-of-field test, based on the near and far bounds from Equation (21), to produce a permissive `foreground_clear` label.

**Discussion.** Across DPDD, Aperture-Dataset, and BLB, the concrete estimation route for $S_1$, sensor size, and aperture differs, yet all three pipelines reduce to the same thin-lens function `calc_dof_cond` and the same reference-width and crop-factor conventions. As a result, every I2I training pair carries a physically grounded and numerically comparable bokeh strength $K$ in its `camera_anns`. This alignment lets Stage-1 share a single bokeh-attention branch across real DSLR photographs, captured depth-assisted triplets, and synthetic renders, while interpreting the requested bokeh strength on a consistent metric scale.

## C. Mathmatical Proofs

**Proposition 4.1** (Depth-from-Bokeh Sweep under Calibrated Bokeh Control). *For a static scene observed by a thin-lens camera with fixed pose and focus distance, we record a bokeh stack by sweeping only the calibrated bokeh strength $K$. At every pixel, the measured bokeh radius is exactly proportional to that pixel's inverse-depth offset from the focal plane. The slope of this proportionality, obtained by regressing radius on $K$ across the stack, is an unbiased and consistent estimate of that offset and yields the pixel's metric depth up to the usual front/behind-focus sign.*

*Proof.* Consider a thin-lens camera of focal length $f$, held at a fixed viewpoint. Let the lens be focused at object distance $S_1 > f$. For any pixel location $x$ in the image plane, let $S_2(x)$ denote the distance from the lens to the 3D scene point that projects to $x$. We define the per-pixel inverse-depth offset from the focal plane,

$$\Delta(x) := \left| \frac{1}{S_1} - \frac{1}{S_2(x)} \right|. \tag{22}$$

We now describe the bokeh sweep acquisition. We capture $m$ images $\{I_{K_i}\}_{i=1}^m$ of the same static scene from the same pose and with the same focus distance $S_1$, while sweeping only a calibrated bokeh strength $K_i$ for each frame $i$. Concretely, frame $i$ is taken with a (aperture) pupil diameter $A_i$ and $f$-number $N_i = f/A_i$, and changing $A_i$ changes $K_i$ but leaves all other camera parameters, including the camera pose and the focus distance $S_1$, fixed. At each setting $i$, the observed image around pixel $x$ is modeled (locally, under a spatially shift-invariant defocus approximation) as a convolution of an all-in-focus radiance image $J$ with an isotropic pillbox PSF of radius $r_i(x)$, plus zero-mean noise $\eta_i(x)$:

$$I_{K_i}(x) = (h_{r_i(x)} * J)(x) + \eta_i(x), \tag{23}$$

where $h_{r_i(x)}$ is a disk PSF of radius $r_i(x)$ in output pixels and $\mathbb{E}[\eta_i(x)] = 0$.

We next express $r_i(x)$ in terms of scene geometry and aperture. Under the paraxial thin-lens model, the circle of confusion (CoC) produced on the sensor by a scene point at distance $S_2(x)$, when the lens is focused at $S_1$, has diameter (in physical sensor-length units)

$$d_i(x) = \frac{|S_2(x) - S_1|}{S_2(x)} \frac{f^2}{N_i (S_1 - f)}. \tag{24}$$

Equation (24) follows directly from similar triangles and the thin-lens relation, and is exact in the paraxial regime for a thin lens with a circular aperture of diameter $A_i = f/N_i$.

We rewrite the geometric factor in (24) using inverse distances. First note that

$$\left| \frac{1}{S_1} - \frac{1}{S_2(x)} \right| = \left| \frac{S_2(x) - S_1}{S_1 S_2(x)} \right| = \frac{|S_2(x) - S_1|}{S_1 S_2(x)}. \tag{25}$$

Multiplying both sides of (25) by $S_1$ gives

$$S_1 \left| \frac{1}{S_1} - \frac{1}{S_2(x)} \right| = \frac{|S_2(x) - S_1|}{S_2(x)}. \tag{26}$$

Substituting (26) into (24) yields

$$d_i(x) = \frac{f^2}{N_i (S_1 - f)} S_1 \left| \frac{1}{S_1} - \frac{1}{S_2(x)} \right| = \frac{f^2 S_1}{N_i (S_1 - f)} \Delta(x), \tag{27}$$

where $\Delta(x)$ is the inverse-depth offset defined in (22).

The pillbox PSF $h_{r_i(x)}$ is parameterized in terms of its radius $r_i(x)$ in *output pixels* rather than physical sensor units. Let pixel_ratio $> 0$ denote the known conversion factor from sensor-length units to output pixels, and recall that $r_i(x)$ is half the CoC diameter measured in pixels. Then

$$r_i(x) = \frac{1}{2} d_i(x) \, \text{pixel\_ratio} = \frac{1}{2} \left[ \frac{f^2 S_1}{N_i (S_1 - f)} \Delta(x) \right] \text{pixel\_ratio}. \tag{28}$$

All terms in brackets in (28) that depend on the aperture index $i$ but not on scene depth at $x$ can be grouped into a known scalar $K_i > 0$, which we call the calibrated defocus gain for aperture $i$:

$$K_i := \frac{1}{2} \frac{f^2 S_1}{N_i (S_1 - f)} \, \text{pixel\_ratio}. \tag{29}$$

This $K_i$ is exactly the per-frame calibrated bokeh strength mentioned in Proposition 4.1: it folds known camera quantities $(f, N_i)$, the fixed focus distance $S_1$, and the pixel scaling factor into a single scalar. By construction, $K_i$ varies only because we deliberately change the aperture for frame $i$, while the scene geometry and $S_1$ stay fixed.

With the calibrated defocus gain for aperture $i$, (28) yields the exact per-pixel linear defocus law

$$r_i(x) = K_i \, \Delta(x) \quad \text{for each } i = 1, \ldots, m. \tag{30}$$

Equation (30) implies that for a fixed pixel $x$, all $\big(K_i, r_i(x)\big)$ pairs fall on a single origin-passing line with slope $\Delta(x)$.

In practice, the "measured bokeh radius" referred to in Proposition 4.1 is obtained from each captured frame rather than taken as the ideal geometric radius $r_i(x)$ itself. We therefore do not observe $r_i(x)$ directly; instead we estimate it from the captured image $I_{K_i}$, e.g. by fitting a pillbox PSF radius. Let $\widehat{r}_i(x)$ denote such a per-frame estimate, and assume it is unbiased with finite variance:

$$\mathbb{E}[\widehat{r}_i(x)] = r_i(x), \qquad \text{Var}[\widehat{r}_i(x)] < \infty. \tag{31}$$

We now form the ordinary least-squares (OLS) slope through the origin that regresses $\widehat{r}_i(x)$ against $K_i$:

$$\widehat{\Delta}(x) := \frac{\sum_{i=1}^{m} K_i \, \widehat{r}_i(x)}{\sum_{i=1}^{m} K_i^2}. \tag{32}$$

To show that $\widehat{\Delta}(x)$ is unbiased for $\Delta(x)$, we first substitute (30) into (31), which implies

$$\mathbb{E}[\widehat{r}_i(x)] = r_i(x) = K_i \, \Delta(x). \tag{33}$$

Taking expectation of (32) and applying (33) termwise in the numerator,

$$\mathbb{E}[\widehat{\Delta}(x)] = \mathbb{E}\left[ \frac{\sum_{i=1}^{m} K_i \, \widehat{r}_i(x)}{\sum_{i=1}^{m} K_i^2} \right] = \frac{\sum_{i=1}^{m} K_i \, \mathbb{E}[\widehat{r}_i(x)]}{\sum_{i=1}^{m} K_i^2} \tag{34}$$

$$= \frac{\sum_{i=1}^{m} K_i \, (K_i \, \Delta(x))}{\sum_{i=1}^{m} K_i^2} = \frac{\Delta(x) \sum_{i=1}^{m} K_i^2}{\sum_{i=1}^{m} K_i^2} = \Delta(x). \tag{35}$$

Hence $\widehat{\Delta}(x)$ is an unbiased estimator of the inverse-depth offset $\Delta(x)$.

We now examine its variance. Assume that the random errors across different aperture settings are uncorrelated, i.e. $\widehat{r}_i(x)$ and $\widehat{r}_j(x)$ are independent for $i \neq j$, and that each $\widehat{r}_i(x)$ has finite variance. Then from (32),

$$\text{Var}[\widehat{\Delta}(x)] = \text{Var}\left[ \frac{\sum_{i=1}^{m} K_i \, \widehat{r}_i(x)}{\sum_{i=1}^{m} K_i^2} \right]$$

$$= \frac{1}{\left( \sum_{i=1}^{m} K_i^2 \right)^2} \sum_{i=1}^{m} K_i^2 \, \text{Var}[\widehat{r}_i(x)]. \tag{36}$$

Because the denominator in (36) grows as $\left( \sum_{i=1}^{m} K_i^2 \right)^2$, the variance decays on the order of $1 / \sum_{i=1}^{m} K_i^2$. In particular, if $\sum_{i=1}^{m} K_i^2 \to \infty$ as $m \to \infty$, then $\text{Var}[\widehat{\Delta}(x)] \to 0$, so $\widehat{\Delta}(x)$ is consistent for $\Delta(x)$.

Finally, we recover metric depth. From (22), we have

$$\Delta(x) = \left| \frac{1}{S_1} - \frac{1}{S_2(x)} \right| \quad \Longrightarrow \quad \frac{1}{S_2(x)} = \frac{1}{S_1} \pm \Delta(x). \tag{37}$$

The $\pm$ corresponds to the front/back ambiguity of defocus: a point in front of the focal plane and a symmetric point behind it yield the same absolute offset $\Delta(x)$. Replacing $\Delta(x)$ by the unbiased, consistent estimator $\widehat{\Delta}(x)$ from (32) gives the per-pixel metric depth estimate

$$\frac{1}{S_2(x)} = \frac{1}{S_1} \pm \widehat{\Delta}(x), \qquad S_2(x) = \left( \frac{1}{S_1} \pm \widehat{\Delta}(x) \right)^{-1}. \tag{38}$$

Taken together, Equations (30), (35), and (38) show that a calibrated sweep of $\{K_i\}_{i=1}^m$ induces a per-pixel linear bokeh–versus–inverse-depth relationship $r_i(x) = K_i \Delta(x)$ whose slope is precisely the inverse-depth offset $\Delta(x)$, and that, under the finite-variance and independence assumptions stated above, the OLS slope $\widehat{\Delta}(x)$ in (32) is an unbiased and consistent estimator of $\Delta(x)$ as $m$ increases (since $\mathrm{Var}[\widehat{\Delta}(x)] \to 0$ when $\sum_i K_i^2 \to \infty$); substituting this estimator into (38) then yields a per-pixel metric depth estimate, subject only to the standard in-front / behind-focus sign ambiguity of defocus.

$\square$

# D. Definition of Evaluation Metrics

We evaluate both stages of our pipeline using established metrics in image synthesis and monocular depth estimation. Below we summarize the definitions adopted in this work.

**Stage-1: Bokeh synthesis quality.** Given a reference bokeh image $B$ and a synthesized bokeh image $\hat{B}$ of size $H \times W$, we first measure distortion with peak signal-to-noise ratio (PSNR) and structural similarity (SSIM). PSNR is defined as

$$\mathrm{PSNR}(B, \hat{B}) = 10 \log_{10} \left( \frac{L^2}{\mathrm{MSE}(B, \hat{B})} \right), \tag{39}$$

where $L$ is the maximum possible pixel value and

$$\mathrm{MSE}(B, \hat{B}) = \frac{1}{HW} \sum_p \left\| B(p) - \hat{B}(p) \right\|_2^2 \tag{40}$$

is the mean squared error over all pixels $p$. SSIM measures local luminance, contrast and structural consistency between $B$ and $\hat{B}$. Following (Wang et al., 2004), we compute the SSIM index over sliding windows as

$$\mathrm{SSIM}(B, \hat{B}) = \frac{\left(2\mu_B \mu_{\hat{B}} + C_1\right)\left(2\sigma_{B\hat{B}} + C_2\right)}{\left(\mu_B^2 + \mu_{\hat{B}}^2 + C_1\right)\left(\sigma_B^2 + \sigma_{\hat{B}}^2 + C_2\right)}, \tag{41}$$

where $\mu_B$ and $\mu_{\hat{B}}$ are local means, $\sigma_B^2$ and $\sigma_{\hat{B}}^2$ are local variances, $\sigma_{B\hat{B}}$ is the local covariance, and $C_1, C_2$ are small constants that stabilize the ratio. Higher PSNR and SSIM indicate better bokeh reconstruction quality.

To better capture perceptual fidelity, we also report LPIPS (Zhang et al., 2018) and DISTS (Ding et al., 2020). LPIPS compares deep features extracted by a pretrained network and averages the channel-wise distance over spatial locations

$$\mathrm{LPIPS}(B, \hat{B}) = \sum_{\ell} w_\ell \frac{1}{|\Omega_\ell|} \sum_{p \in \Omega_\ell} \left\| \phi_\ell(B)_p - \phi_\ell(\hat{B})_p \right\|_2^2, \tag{42}$$

where $\phi_\ell(\cdot)$ denotes features at layer $\ell$, $\Omega_\ell$ is the corresponding spatial grid, and $w_\ell$ are learned scalar weights. DISTS computes a weighted combination of structure and texture similarity in deep feature space

$$\mathrm{DISTS}(B, \hat{B}) = \sum_{\ell} \left[ \alpha_\ell \left(1 - S_\ell(B, \hat{B})\right) + \beta_\ell \left(1 - T_\ell(B, \hat{B})\right) \right], \tag{43}$$

where $S_\ell$ measures structural similarity between normalized features, $T_\ell$ measures texture similarity through feature magnitudes, and $\alpha_\ell, \beta_\ell$ are learned nonnegative weights. Lower LPIPS and DISTS indicate that synthesized bokeh images are closer to the reference in a feature space that correlates with human perception. All image quality metrics are computed on linear RGB images, cropped to the valid field of view of each dataset.

**Stage-2: Metric depth estimation.** For Stage-2, we evaluate predicted metric depth maps against ground-truth depth using standard monocular depth metrics. Let $D$ be the predicted depth map and $D^*$ the ground-truth depth, defined on the set of valid pixels $\mathcal{V}$. The absolute relative error (AbsRel) is defined as

$$\text{AbsRel} = \frac{1}{|\mathcal{V}|} \sum_{p \in \mathcal{V}} \frac{|D(p) - D^*(p)|}{D^*(p)}. \tag{44}$$

We treat lower AbsRel as better and report it as our main scalar error metric in tables and ablations.

We also report the common threshold accuracy $\delta_1$, which measures the fraction of pixels where the prediction is close to the ground truth up to a multiplicative factor

$$\delta_1 = \frac{1}{|\mathcal{V}|} \left| \left\{ p \in \mathcal{V} \ \middle| \ \max\left( \frac{D(p)}{D^*(p)}, \frac{D^*(p)}{D(p)} \right) < 1.25 \right\} \right|. \tag{45}$$

Higher $\delta_1$ indicates better agreement between predicted and true depths. For completeness, we also monitor squared relative error (SqRel), root mean squared error (RMSE), and RMSE in log space on the validation sets. Following standard depth-estimation benchmarks, SqRel for a prediction $\hat{D}$ and ground-truth depth $D$ over pixels $\Omega$ is defined as

$$\text{SqRel} = \frac{1}{|\Omega|} \sum_{p \in \Omega} \frac{\left(D_p - \hat{D}_p\right)^2}{D_p}, \tag{46}$$

and RMSE is defined as

$$\text{RMSE} = \sqrt{\frac{1}{|\Omega|} \sum_{p \in \Omega} \left(D_p - \hat{D}_p\right)^2}, \tag{47}$$

with $\text{RMSE}_{\log}$ computed analogously in log space. We omit some of these metrics from the main tables when their trends are consistent with the primary ones. All depth metrics are computed in metric units on the valid depth range of each dataset without scale alignment, following the standard protocol used in recent monocular metric depth work.

## E. Details of *SYNTHEBOKEH300* Dataset

*SYNTHEBOKEH300*, shown in Figure 7, is a synthetic benchmark that exposes fine-grained control over defocus strength and focal distance under fully known geometry. The dataset is built on a two-layer multi-plane image representation rendered with a ray-traced thin-lens model. For each scene we provide a sharp all-in-focus image, a floating point disparity map and multiple bokeh images rendered under different aperture and focus settings. All images are RGB at a resolution of $1024 \times 1024$.

We construct each scene by compositing a foreground RGBA matte over a natural background photograph. Foreground assets are PNG images with transparency that capture objects with complex silhouettes such as people, plants and everyday items. The generator first crops the foreground to the tight alpha bounding box, then randomly rescales it so that the projected area occupies roughly 30%–80% of the final frame. The resized foreground is placed near the image center with a small random offset while we ensure that it remains fully inside the background canvas. Background images are resized to a slightly larger canvas than the final resolution to absorb boundary effects introduced by lens blur. After rendering we crop a central $1024 \times 1024$ window which defines a single all-in-focus reference image per foreground–background pair.

Depth in *SYNTHEBOKEH300* is defined in disparity space to match the thin-lens formulation used for camera calibration in our real-image datasets. For the background we sample a random planar disparity field

$$d_{\text{bg}}(x, y) = \frac{c}{1 - ax - by}, \tag{48}$$

where $(a, b, c)$ are normalized so that $d_{\mathrm{bg}}$ remains within a bounded range over the image grid. The foreground is assigned a separate planar disparity band whose values lie closer to the camera than the background disparity within the alpha support, ensuring smooth disparity in both layers and a consistent occlusion order. We store the final per-pixel disparity map $d(x, y)$ for each scene as a single-channel `float32` array, providing ground-truth metric depth up to a global scale factor.

Given the layered scene representation we render defocus using a reverse ray-tracing module. For each scene the renderer takes as input the linear RGB foreground and background layers, their opacity masks and the corresponding disparity coefficients and simulates a thin lens with a finite aperture. We parameterize defocus by a dimensionless blur strength $K$ and a normalized focus disparity $d_f \in [0, 1]$. The renderer uses $K$ to scale the circle-of-confusion radius in disparity space as

$$\Delta d(x, y) = K \frac{d(x, y) - d_f}{s}, \tag{49}$$

with $s$ a fixed defocus scale factor. The module integrates multiple rays per pixel to obtain a bokeh image in linear RGB, which is then converted back to display gamma with exponent $1/\gamma$ with $\gamma = 2.2$.

For each foreground–background combination we keep the geometry fixed and vary only the lens parameters. The generator samples $K$ from a wide range that spans both subtle and strong blur, and restricts $d_f$ to the foreground-focused regime so that synthesized views keep the main subject sharp while varying the background blur. In our default configuration we draw three values of $K$ and one value of $d_f$, which yields three distinct bokeh renderings per scene while sharing a single all-in-focus image and disparity map. We also map each sampled $K$ to an equivalent $f$-number in the range $[1.4, 22]$ in order to align the synthetic lens settings with typical DSLR cameras. The sampling policy is encoded in the metadata and matches the distribution of lens parameters used when training the editing model on real-camera datasets.

The final dataset is organized under four top-level directories: `aif` for all-in-focus inputs, `images` for bokeh renderings, `depth` for disparity maps and `metadata` for per-image JSON descriptors. Each bokeh image is stored as an 8-bit JPEG in `images` and has a corresponding JSON file in `metadata` that records its identifier, the shared all-in-focus and depth file paths, the sampled $K$ and $d_f$, the equivalent $f$-number and basic renderer settings.

During evaluation *SYNTHEBOKEH300* provides photorealistic yet fully controllable test cases in which we can measure both image reconstruction quality and depth-aware consistency across changes in aperture and focus under ground-truth geometry.

# F. Other Ablation Studies

### F.1. Ablation on Stage-1 Inference Steps

To understand how the number of reverse diffusion steps in Stage-1 affects bokeh synthesis quality, we evaluate ***BokehDepth*** on the SYNTHEBOKEH300 validation set under different numbers of inference steps. Table 7 reports standard distortion and perceptual metrics for 50, 10, 5, and 1 sampling steps.

| Steps | PSNR↑ | SSIM↑ | LPIPS↓ | DISTS↓ |
|---|---|---|---|---|
| 50 | 29.1215 | 0.9011 | 0.1385 | 0.0725 |
| 10 | 27.8744 | 0.8900 | 0.1444 | 0.0751 |
| 5 | 28.2572 | 0.8829 | 0.1584 | 0.0824 |
| 1 | 24.8304 | 0.8539 | 0.1671 | 0.0946 |

*Table 7.* Ablation on the number of Stage-1 inference steps on the SYNTHEBOKEH300 validation set.

Using 50 steps yields the best overall reconstruction quality, but the gap between 50 and 10 steps is modest in both PSNR and SSIM, and the perceptual metrics remain close. Comparing with the results in Figure 10, reducing the step count further to 5 or 1 leads to clear degradation across almost all metrics, which indicates insufficient convergence of the diffusion sampler and more noticeable artifacts in the synthesized bokeh.

Considering the cost of generating a full bokeh stack for Stage-2, we adopt 10 diffusion steps for all Stage-1 bokeh stack generation. This configuration achieves a balanced trade off between fidelity and efficiency and keeps the training and evaluation pipeline computationally practical.

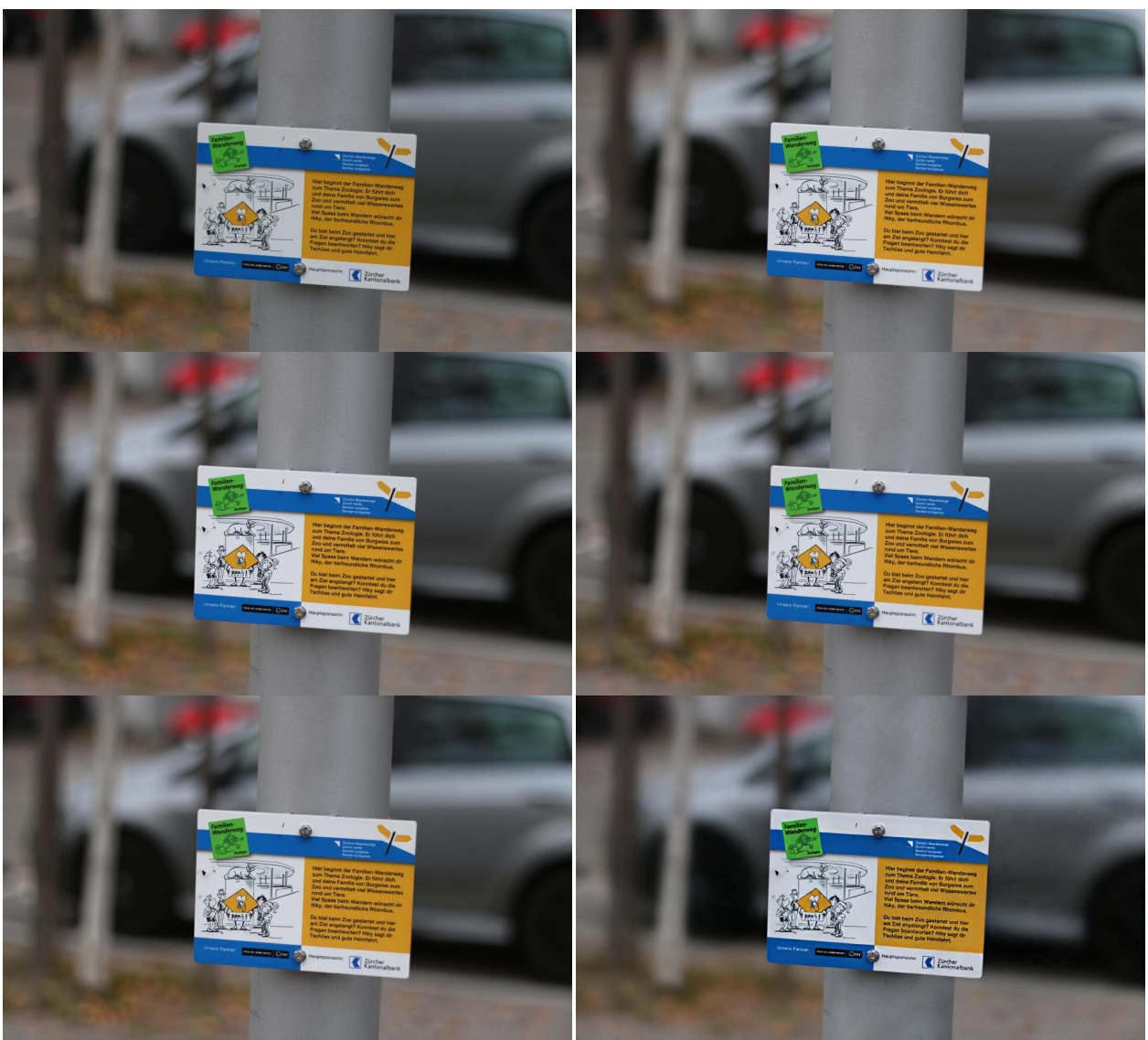

*Figure 10.* Qualitative results of the ablation on the number of Stage-1 inference steps. From left to right and top to bottom we show the ground-truth bokeh image, followed by **BokehDepth** results with 50, 20, 10, 5, and 1 inference step on the same scene.

*Table 8.* Ablation on stack size $N$ and blur range $K$ (KITTI Eigen split). Best in **bold**, second best underlined.

| $N$ | $K$ range | $\delta_1\uparrow$ | $\delta_2\uparrow$ | $\delta_3\uparrow$ | AbsRel$\downarrow$ | SqRel$\downarrow$ | RMSE$\downarrow$ | RMSE$_{\log}\downarrow$ | $\log_{10}\downarrow$ | SiLog$\downarrow$ |
|---|---|---|---|---|---|---|---|---|---|---|
| 1 | $[10, 20]$ | 0.918 | 0.987 | 0.997 | 0.0950 | 0.370 | 3.029 | 0.1291 | 0.0401 | 0.1252 |
| 1 | $[20, 30]$ | 0.918 | 0.987 | 0.997 | 0.0951 | 0.370 | 3.026 | 0.1291 | 0.0400 | 0.1249 |
| 2 | $[10, 20]$ | 0.928 | 0.988 | 0.997 | 0.0860 | 0.342 | 2.881 | 0.1200 | 0.0375 | 0.1250 |
| 2 | $[20, 30]$ | 0.928 | 0.988 | 0.997 | 0.0859 | 0.341 | 2.875 | 0.1200 | 0.0374 | 0.1249 |
| 2 | $[10, 30]$ | 0.932 | 0.989 | 0.997 | 0.0854 | 0.332 | 2.846 | 0.1189 | 0.0374 | 0.1224 |
| 3 | $[10, 30]$ | 0.943 | 0.992 | 0.998 | 0.0842 | 0.296 | 2.725 | 0.1163 | 0.0374 | 0.1142 |
| 5 | $[10, 30]$ | **0.943** | **0.993** | **0.998** | **0.0838** | **0.290** | **2.691** | **0.1156** | **0.0370** | **0.1130** |

*Table 9.* Inference cost *vs.* stack size $N$ ($K \in [10, 30]$, Stage-1 uses 30 diffusion steps).

| $N$ | Time (s) | | | Memory (GB) | |
|---|---|---|---|---|---|
| | Stage-1 | Stage-2 | Total | Stage-1 | Stage-2 |
| 1 | 16.76 | 0.08 | 16.84 | 33.08 | 26.20 |
| 2 | 32.54 | 0.10 | 32.64 | 33.70 | 26.23 |
| 3 | 48.66 | 0.12 | 48.78 | 34.32 | 26.33 |
| 4 | 64.46 | 0.14 | 64.60 | 34.94 | 26.53 |
| 5 | 79.50 | 0.16 | 79.66 | 35.56 | 26.80 |

## F.2. Ablation on Bokeh Stack Size and Blur Range

We ablate two key hyperparameters of the defocus stack: the number of rendered bokeh frames $N$ and the blur-kernel range $K \in [K_{\min}, K_{\max}]$. All variants are evaluated on the KITTI Eigen split under the same Stage-2 setting as Table 5.

**Depth accuracy.** Table 8 reports results across configurations. Two trends emerge. *(i) More frames help, with diminishing returns.* A single defocused frame ($N{=}1$) is clearly insufficient; moving to $N{=}2$ yields a substantial gain (e.g., $\delta_1$ improves from 0.918 to 0.932), and $N{=}3$ improves further ($\delta_1{=}0.943$). However, the marginal gain from $N{=}3$ to $N{=}5$ is small ($\delta_1$: 0.943→0.943, AbsRel: 0.0842→0.0838). *(ii) Spread of $K$ matters more than its absolute location.* Under fixed $N{=}2$ with a narrow range, $K{\in}[10, 20]$ and $K{\in}[20, 30]$ perform almost identically, while the wider range $K{\in}[10, 30]$ consistently outperforms both (e.g., RMSE drops from 2.881 to 2.846). This indicates that the effective spread of the calibrated defocus sweep, rather than simply shifting toward larger blur kernels, is the primary driver of accuracy.

Both observations align with Proposition 4.1, which models depth recovery as a per-pixel regression on $K$. Adding observations (larger $N$) and increasing their spread (wider $[K_{\min}, K_{\max}]$) improve the conditioning of this regression, reducing the OLS variance at rate $\mathcal{O}\!\left(\frac{1}{N \operatorname{Var}(K)}\right)$ (Equation (36)).

**Inference cost.** Table 9 profiles the wall-clock time and peak GPU memory as a function of $N$ (Stage-1 uses 30 diffusion steps). Stage-1 dominates the runtime and scales nearly linearly with $N$, while Stage-2 adds negligible overhead ($< 0.2\,$s for all $N$). Memory grows modestly with $N$ due to the additional latent caching in Stage-1. Given that accuracy largely saturates after $N{=}3$ while cost continues to grow linearly, we adopt $N{=}3$ with $K{\in}[10, 30]$ as the default configuration throughout all other experiments.

## F.3. Ablation on Defocus Cues and Generator Artifacts

We further analyze whether the proposed DSFA module learns physically meaningful defocus cues or merely exploits appearance artifacts introduced by the Stage-1 generator. To disentangle these factors, we expand the Stage-2 ablation in Table 10. All variants are evaluated on the KITTI Eigen split under the same setting as Table 5. We compare five configurations: the monocular depth baseline, DSFA fed with BokehMe renderings from sparse LiDAR depth, DSFA fed with BokehMe renderings from dense predicted depth, DSFA fed with raw FLUX.1-Kontext outputs, and our full calibrated Stage-1 bokeh stack.

The results lead to two observations. First, multiple bokeh patterns are useful only when the rendered defocus stack is

*Table 10.* Extended Stage-2 ablation on the KITTI Eigen split. The results show that DSFA benefits from bokeh stacks only when the defocus cues are dense, spatially coherent, and physically calibrated.

| Method | $\delta_1 \uparrow$ | $\delta_2 \uparrow$ | $\delta_3 \uparrow$ | AbsRel $\downarrow$ | SqRel $\downarrow$ | RMSE $\downarrow$ | RMSE(log) $\downarrow$ | log10 $\downarrow$ | SiLog $\downarrow$ |
|---|---|---|---|---|---|---|---|---|---|
| DAv2 | 0.914 | 0.987 | 0.997 | 0.097 | 0.379 | 3.030 | 0.130 | 0.041 | 0.130 |
| GT-Sparse-Depth + BokehMe + DSFA | 0.914 | 0.988 | 0.997 | 0.095 | 0.378 | 3.030 | 0.129 | 0.040 | 0.129 |
| Pred-Dense-Depth + BokehMe + DSFA | 0.918 | 0.989 | 0.997 | 0.094 | 0.370 | 3.022 | 0.128 | 0.040 | 0.125 |
| FLUX.1-Kontext + DSFA | 0.608 | 0.884 | 0.958 | 0.227 | 1.387 | 5.860 | 0.477 | 0.112 | 0.475 |
| Ours (Stage-1 + DSFA) | **0.943** | **0.992** | **0.998** | **0.084** | **0.296** | **2.725** | **0.116** | **0.037** | **0.114** |

spatially usable. The variant using BokehMe with sparse LiDAR depth improves only marginally over the no-bokeh baseline, from 0.097 to 0.095 in AbsRel. This does not imply that defocus cues are ineffective. Instead, it reflects a limitation of using sparse KITTI LiDAR samples as the rendering geometry. Sparse depth provides incomplete per-pixel structure, especially around object boundaries and thin regions, where defocus rendering is most sensitive. As a result, the rendered stack contains unstable blur boundaries and limited spatial continuity, making it difficult for DSFA to extract reliable defocus-to-depth evidence. When the sparse depth input is replaced with a dense predicted depth map, the performance improves consistently, reaching 0.094 AbsRel and 0.125 SiLog. This indicates that DSFA can benefit from multiple bokeh patterns, but the stack must provide dense and coherent spatial cues.

Second, multiple generated bokeh images alone are not sufficient. If DSFA were mainly exploiting visual artifacts or style biases from a generative backbone, then feeding it raw FLUX.1-Kontext bokeh outputs should still provide useful cues. However, this variant performs substantially worse, with AbsRel increasing to 0.227 and $\delta_1$ dropping to 0.608. This failure shows that DSFA does not benefit from arbitrary generated blur patterns. Rather, it requires the blur changes across the stack to follow a consistent and calibrated defocus axis. This is the key distinction of our Stage-1 design: instead of producing unrelated stylized bokeh images, it synthesizes a multi-strength bokeh stack whose blur variation is explicitly controlled by the physical parameter $K$. The DSFA module can therefore aggregate features along a meaningful defocus sweep, rather than over unconstrained appearance changes.

Overall, this ablation suggests that the gain of our method is not caused by overfitting to FLUX.1-Kontext artifacts. The decisive factor is whether the bokeh stack is dense, cross-frame consistent, and physically calibrated. BokehMe with sparse LiDAR depth satisfies this condition only partially, dense-depth BokehMe improves the spatial usability of the stack, raw FLUX.1-Kontext satisfies none of these requirements, and our Stage-1 satisfies all three. This explains why the full model achieves the best performance, improving AbsRel from 0.097 to 0.084 and SiLog from 0.130 to 0.114 over the DAv2 baseline.

## G. Stage-2 Depth Range analysis

We aggregate errors over all samples and all valid pixels to perform a pixel-weighted global analysis. This evaluation quantifies how much DSFA improves over the baseline in overall accuracy, and it also reveals where the gains concentrate across different depth ranges and around image edges. We report two standard regression metrics, mean absolute error (MAE) and root mean squared error (RMSE), both computed on metric depth values. For each valid pixel $p$, we measure the absolute error as $e_p = |d_p - \hat{d}_p|$ and the squared error as $(d_p - \hat{d}_p)^2$, where $d_p$ is the ground-truth depth and $\hat{d}_p$ is the predicted depth. Global MAE and RMSE are then obtained by summing these per-pixel errors across all images and dividing by the total number of valid pixels. In addition, we report the improved-pixel fraction, defined as the percentage of valid pixels whose absolute error decreases under DSFA, namely $\mathbb{1}[e_p^{\text{base}} - e_p^{\text{DSFA}} > \tau]$ with threshold $\tau$. To probe boundary quality, we compute an edge mask from the RGB gradient and evaluate MAE on the edge subset only. Finally, we provide depth-stratified MAE by grouping pixels into bins based on ground-truth depth and averaging the absolute error within each bin, which helps diagnose whether improvements come from near-range geometry, far-range structure, or both.

As shown in Table 11, DSFA delivers a substantial and consistent improvement over the baseline. Importantly, the gain is not limited to global metrics such as MAE and RMSE, but also extends to the more challenging edge pixels. These edge regions often correspond to occlusion boundaries and thin structures, which are particularly vulnerable to the over-smoothing bias of monocular priors. DSFA achieves a clear reduction in edge MAE, indicating that it effectively leverages the additional cues to correct boundary geometry rather than merely improving easy interior areas.

The depth-range analysis in Table 12 further supports this conclusion. DSFA yields positive improvements across all

*Table 11.* Depth Anything V2 (Yang et al., 2024b) results on IBims-1 (Koch et al., 2018) before and after DSFA. Gains are computed as **Baseline − DSFA**, so positive values indicate improvement.

| Split | Metric | Baseline | DSFA | Gain | Count |
|-------|--------|---------:|-----:|-----:|------:|
| Global | MAE | 0.3518 | 0.1925 | +0.1593 | 29,293,761 |
| Global | RMSE | 0.6097 | 0.3559 | +0.2538 | 29,293,761 |
| Global | Improved pixel fraction | – | 0.6565 | – | 29,293,761 |
| Edges | MAE | 0.4376 | 0.2510 | +0.1866 | 4,297,906 |

*Table 12.* Depth Anything V2 (Yang et al., 2024b)'s Depth-stratified MAE on IBims-1 (Koch et al., 2018). Each bin reports the pixel-weighted MAE computed over pixels whose ground-truth depth falls within the specified range.

| Depth range | Baseline MAE | DSFA MAE | Gain (Base − DSFA) | Valid pixels |
|-------------|-------------:|---------:|-------------------:|-------------:|
| $[0, 0.5)$ | 1.9363 | 1.4429 | +0.4934 | 509 |
| $[0.5, 1)$ | 0.2033 | 0.1065 | +0.0968 | 618,520 |
| $[1, 2)$ | 0.1985 | 0.1082 | +0.0902 | 7,940,167 |
| $[2, 4)$ | 0.3120 | 0.1624 | +0.1496 | 14,546,559 |
| $[4, 6)$ | 0.5294 | 0.3031 | +0.2263 | 4,183,729 |
| $[6, 10)$ | 0.9227 | 0.5401 | +0.3826 | 2,004,102 |
| $[10, +\infty)$ | 2.0448 | 0.7526 | +1.2922 | 175 |

depth intervals, and the margin becomes more pronounced at larger depths. This trend aligns with the intuition that monocular depth estimation increasingly relies on semantic scale assumptions in the far range, while focal stacks provide complementary and directly observable constraints. Taken together, these results suggest that DSFA offers a systematic improvement, affecting a large fraction of pixels and enhancing boundary fidelity, instead of reflecting incidental fluctuations in averaged scores.

## H. Additional Qualitative Results

We provide additional qualitative results for Stage-1 in Figure 8 and Figure 9, and for the full pipeline in Figure 6, Figure 11, and Figure 12.

We visualize DSFA improvements using two complementary maps. The $\Delta$Error map is a diverging visualization of the error difference. Given $|\text{err}_{\text{base}}| = |\text{pred}_{\text{base}} - \text{gt}|$ and $|\text{err}_{\text{dsfa}}| = |\text{pred}_{\text{dsfa}} - \text{gt}|$, we define $\Delta = |\text{err}_{\text{base}}| - |\text{err}_{\text{dsfa}}|$. Red regions indicate larger baseline errors and smaller DSFA errors, which means DSFA improves prediction accuracy in those areas.

For the overlay visualization, we define an improvement mask as $\text{improvement\_mask} = (\Delta > 0.001) \wedge \text{mask}$. Pixels satisfying this condition are rendered in green on top of the original RGB image. Green regions indicate that DSFA strictly reduces the absolute error by at least 0.001 meters at those pixels.

## I. Limitations.

The core evaluation of our method focuses on single-image monocular metric depth estimation. However, real-world deployment often requires video-level temporal consistency, robustness to dynamic scenes with moving objects, and stable performance under practical imaging factors such as exposure variation, rolling shutter, and focus drift. In addition, our bokeh stack is produced by editing a single viewpoint into multiple defocus levels, which does not fully reflect the temporal imaging process of a real camera. As a result, it remains unclear whether the proposed defocus cues can consistently improve depth estimation in video settings, and validating this extension is an important direction for future work.

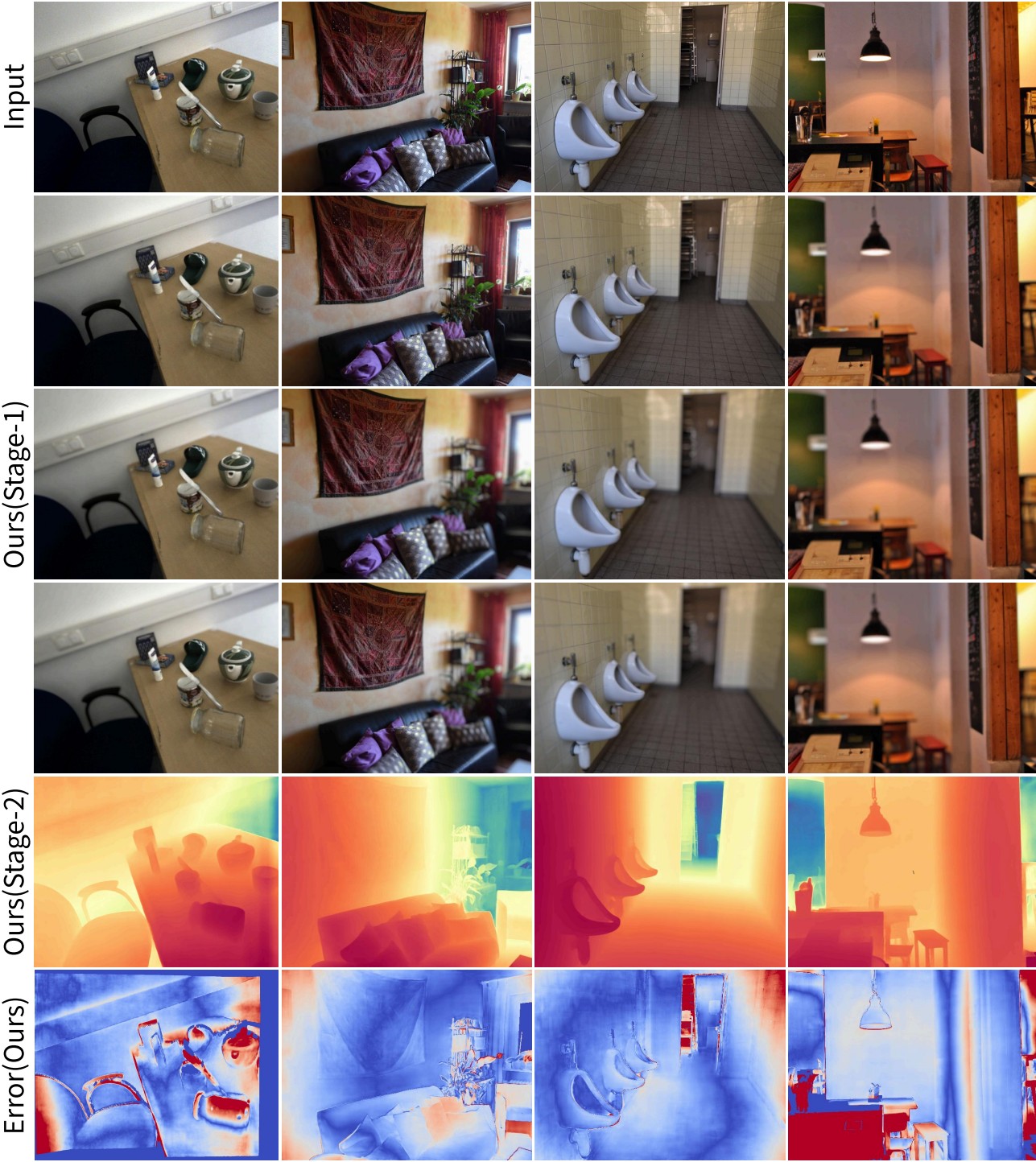

*Figure 11.* Qualitative results of ***BokehDepth*** using the Depth Anything V2 ([Yang et al., 2024b](#)) backbone. From top to bottom: the input image, three representative frames from the Stage-1 bokeh stack, the Stage-2 depth prediction, and the error map of **BokehDepth** .

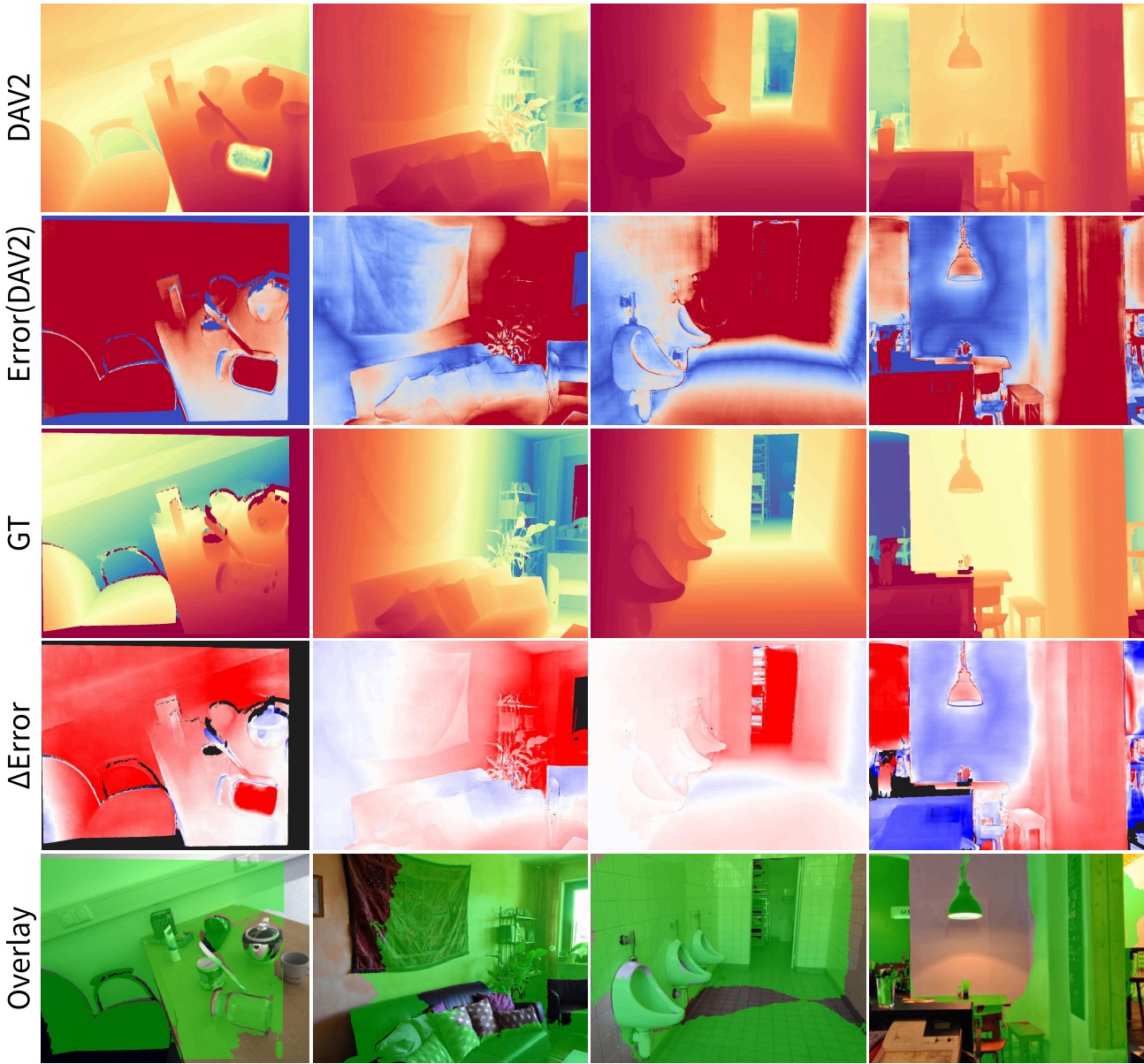

*Figure 12.* Qualitative results of ***BokehDepth*** using the Depth Anything V2 (Yang et al., 2024b) backbone (continued from Figure 11). From top to bottom: the Depth Anything V2 prediction, the corresponding error map, the ground truth depth, the ΔError map that reports the per-pixel reduction in absolute depth error of **BokehDepth** over the base model, and the RGB image overlaid with green regions that mark where our method produces notable improvements. **BokehDepth** lowers depth errors on fine structures, weakly-textured walls and distant background regions, offering more distinct layer separation and steadier metric depth across varied scenes.

