# OpenReview forum: "Boosting Monocular Metric Depth Estimation via Bokeh Rendering"
_ICML.cc/2026/Conference — ICML 2026 regular_

### Official Review · Reviewer_oyeW · 2026-02-24

**Soundness:** 3
**Presentation:** 3
**Significance:** 2
**Originality:** 3
**Overall Recommendation:** 4
**Confidence:** 4

**Summary:**

BokehDepth proposes a two-stage framework that reverses the traditional relationship between bokeh rendering and depth estimation. While conventional pipelines use predicted depth maps to render bokeh, this work leverages synthetic defocus as a supervision-free geometric signal to improve monocular metric depth estimation.

Stage-1 (Bokeh Generation): The authors build a physically grounded bokeh generator on top of the FLUX-Kontext image-editing backbone. By mapping various optical parameters to a single thin-lens-derived control scalar $K$, the model generates calibrated bokeh stacks from a single sharp image without requiring prior depth information.

Stage-2 (Depth Estimation): These bokeh stacks are integrated into a standard monocular depth encoder using a lightweight Divided Space Focus (DSF) Attention module. This allows the model to extract consistent geometric features from defocus variations, particularly in weakly textured or distant regions where standard models struggle.

Experimental results demonstrate that BokehDepth achieves superior visual fidelity in bokeh synthesis and consistently enhances the metric accuracy of state-of-the-art depth models across diverse indoor and outdoor benchmarks.

**Compliance With Llm Reviewing Policy:**

Affirmed.

**Key Questions For Authors:**

Please check my weakness. My concern is majorly about the fairness in evaluation. I might decrease my score after the rebuttal if this concern is not fully addressed.

**Limitations:**

yes

**Strengths And Weaknesses:**

- Pros

     The paper presents an interesting conceptual shift by decoupling bokeh synthesis from depth prediction. It effectively demonstrates that the reciprocity between defocus and depth can be exploited in reverse—using a generative "bokeh stack" to supervise and refine metric depth estimation.

    The framework achieves competitive or superior results on both tasks. It consistently boosts the performance of strong foundation models (e.g., Depth Anything V2).

    Qualitatively, the method produces depth maps with cleaner occlusion boundaries and better layer separation in challenging scenarios.

    Comprehensive Evaluation: The ablation studies effectively validate the necessity of each component, such as the Focus and Space branches within the DSF Attention module.

- Cons

    Architectural Fairness in Comparisons: A primary concern is the potential unfair issue in the bokeh synthesis evaluation. Stage-1 utilizes the powerful FLUX-Kontext backbone, whereas key baselines like BokehDiff are based on older architectures like SDXL. This makes it difficult to evaluate the proposed method in the Bokeh rendering feild.

    While the bokeh-to-depth idea is interesting, the overall technical contribution is limited.

    Efficiency and Practicality Concerns: The framework lacks a detailed inference time and complexity analysis. Given that Stage-1 requires a multi-step diffusion process to generate a stack of images before depth estimation can even begin, the computational overhead is likely significantly higher than standard single-pass metric depth models. This may limit the method's practicality for real-time applications.

---

> ### Author Rebuttal · Authors · 2026-03-31
>
> Thanks for your thoughtful review.
>
> ## Response to C1&Q
>
> Leveraging a real-image editing model to perform bokeh rendering on real images is itself an important part of our contribution. Following the reviewer's suggestion, we add two recent FLUX.1-based bokeh rendering methods, DiffCamera [1] and GenFocus [2], to Table 1 under identical evaluation settings:
>
> | Method | Backbone | PSNR↑ | SSIM↑ | LPIPS↓ | DISTS↓ | PSNR↑ | SSIM↑ | LPIPS↓ | DISTS↓ |
> |:---|:---|:---:|:---:|:---:|:---:|:---:|:---:|:---:|:---:|
> | | | *EBB! Val200 (real)* | | | | *SYNTHEBOKEH300 (synthetic)* | | | |
> | BokehMe | Classical | 24.13 | 0.751 | 0.390 | 0.143 | 22.83 | 0.809 | 0.279 | 0.188 |
> | DrBokeh | Classical | 22.61 | 0.735 | 0.435 | 0.178 | ***29.70*** | **0.895** | **0.149** | **0.087** |
> | BokehDiff | SDXL | 24.35 | 0.802 | 0.279 | 0.112 | 26.21 | 0.807 | 0.288 | 0.142 |
> | FLUX-Kontext | FLUX | 19.92 | 0.645 | ***0.165*** | 0.427 | 20.34 | 0.685 | 0.351 | 0.155 |
> | DiffCamera [1] | FLUX | 25.07 | **0.867** | 0.311 | 0.102 | 25.26 | 0.880 | 0.238 | 0.120 |
> | GenFocus [2] | FLUX | **25.48** | ***0.886*** | 0.278 | **0.081** | 21.54 | **0.895** | 0.317 | 0.153 |
> | **Ours** | FLUX | ***25.91*** | 0.856 | **0.185** | ***0.076*** | **29.12** | ***0.901*** | ***0.139*** | ***0.073*** |
>
> Our method outperforms both FLUX-based competitors on the perceptual metrics (LPIPS, DISTS)  and leads or matches on PSNR/SSIM. The comparison is now controlled for backbone capacity, and the advantage persists.
>
> [1] DiffCamera: Arbitrary Refocusing on Images, SIGGRAPH Asia 2025
>
> [2] Generative Refocusing: Flexible Defocus Control from a Single Image, arXiv 2026
>
> ## Response to C2
>
> We respectfully disagree that the contribution is limited. BokehDepth introduces a novel research direction by leveraging physically calibrated defocus as a supervision-free geometric cue for monocular metric depth. The technical contributions span two stages. Stage-1 contributes a Bokeh Cross-Attention mechanism that turns a pretrained image editor into a calibrated, depth-free bokeh renderer, as validated by the backbone-controlled comparison above. Stage-2 contributes DSFA, a plug-and-play encoder module that injects multi-frame defocus cues into any existing depth backbone without modifying its decoder or loss. Both components are novel and independently evaluated. The combined framework improves multiple strong baselines across multiple benchmarks in both in-domain and zero-shot settings.
>
> ## Response to C3
>
> **The profiling details are provided in our response to Reviewer xxtu (Response to W1)**. Our core module for depth estimation, DSFA (Stage-2), introduces minimal overhead. We acknowledge the current full-scale diffusion backbone prohibits real-time Stage-1 execution. However, applying well-established one-step or few-step distillation techniques offers a direct optimization path. We will explicitly discuss this future trajectory in the final manuscript to clarify that the current latency is a solvable engineering challenge rather than a fundamental methodological flaw.

---

> > ### Author Rebuttal · Reviewer_oyeW · 2026-04-02
> >
> > All my concerns are addressed and I would like to keep my original score.

---

> > > ### Author Response · Authors · 2026-04-02
> > >
> > > We sincerely thank you for acknowledging our response and for your continued support of our work. We are glad that your concerns have been fully addressed. Thank you again for your time and constructive review!

---

### Official Review · Reviewer_MY8A · 2026-03-12

**Soundness:** 3
**Presentation:** 2
**Significance:** 2
**Originality:** 3
**Overall Recommendation:** 4
**Confidence:** 4

**Summary:**

The paper proposes a two-stage framework that exploits the physical relationship between lens defocus and scene geometry to improve monocular metric depth estimation. In Stage-1, a FLUX-Kontext-based image editing model is adapted with a bokeh cross-attention module conditioned on a thin-lens-derived scalar K, producing calibrated multi-strength bokeh stacks from a single sharp image. In Stage-2, a Divided Space Focus Attention (DSFA) module is inserted into a ViT-based depth encoder, fusing features along the defocus axis to expose depth-sensitive blur variations while the downstream DPT decoder and metric head remain unchanged.

**Compliance With Llm Reviewing Policy:**

Affirmed.

**Final Justification:**

My concerns have been addressed.

**Key Questions For Authors:**

- The BokehDepth (w/ BokehMe) ablation in Table 4 feeds DSFA with bokeh rendered by BokehMe from ground-truth depth, achieves $\delta$ = 0.914 and AbsRel = 0.094, nearly identical to the no-bokeh baseline $\delta$ = 0.914, AbsRel = 0.097. While Stage-1 outputs achieve$\delta$ = 0.943 and AbsRel = 0.084. Since BokehMe with ground-truth depth produces geometrically accurate bokeh with correct per-pixel blur, it should provide DSFA with a stronger and more reliable defocus signal than Stage-1's generative outputs. The large gap between these two variants suggests that DSFA is not learning a generalizable defocus-to-depth mapping from the bokeh stack, but rather exploiting appearance-specific properties of Stage-1-generated images. Since Stage-2 is trained exclusively on Stage-1 outputs, is DSFA simply overfitted to the generative artifacts of FLUX-Kontext rather than learning physically meaningful defocus cues?

**Limitations:**

The paper does not provide limitation discussion on main paper.

**Strengths And Weaknesses:**

**Strength**
- The intuition is interesting and well-motivated. Rather than predicting depth first and rendering bokeh from it, BokehDepth generates physically grounded bokeh stacks first, and uses those stacks to improve depth prediction.

- Table 3 reports proposed method achieving RMS = 0.043 on NYUv2, compared to DepthAnythingV2 = 0.206 and UniDepthV2 = 0.180. The proposed method achieves more than **x4 improvement** on a well-saturated benchmark.

**Weakness**
- The central objective of the paper is ambiguously. The title clearly states the paper's objective is to improve depth estimation by leveraging bokeh rendering. Stage-1 serves as a prerequisite component for Stage-2, and the primary evaluation should be on depth accuracy. To justify this pipeline, the critical experiment is to show that better Stage-1 bokeh quality leads to better Stage-2 depth accuracy. However, Stage-1 alone occupies half content of the method section with its own evaluation and benchmark. By contrast, Stage-2, which is the paper's primary contribution per its title, receives less methodological elaboration and no analysis connecting Stage-1 quality to depth accuracy. The ablation study, BokehDepth (w/ BokehMe) in Table 4, shows that substituting different bokeh products yields no depth improvement over the no-bokeh baseline. It remains unclear whether Stage-1 quality matters at all for Stage-2.

- The Stage-2 training details are not provided. Section 4 describes the DSFA architecture but does not provide the loss function used to train Stage-2. It is stated only that the model is fine-tuned using each base model's published training pipeline. It is unclear whether the defocus cue actually contributes to metric supervision, or whether Stage-2 is simply a domain adaptation finetuning that can improve regardless of whether the bokeh stack is physically calibrated.

---

> ### Author Rebuttal · Authors · 2026-03-31
>
> Thanks for your thoughtful review.
> ## Response to W1
> We clarify that BokehDepth is a two-stage framework: Stage-1 solves depth-free calibrated bokeh synthesis, and Stage-2 is a lightweight plug-in that injects these cues into standard depth backbones. Stage-1 receives more methodological detail because reliable depth-free bokeh synthesis is the key enabler of the pipeline. Moreover, the expanded Table 4 shows that better Stage-1 bokeh leads to better Stage-2 depth. Please see Response to Q for details.
>
> ## Response to W2
>
> Stage-2 strictly preserves the original loss functions and decoders of each backbone, which justifies the generality of our method. Specifically, DAv2 utilizes the standard Scale-Invariant Log loss. UniDepthV2 retains its native composite loss comprising depth, camera, invariance, scale-shift invariant, and confidence terms. Given strict rebuttal character limits, we respectfully direct the reviewers to the original DAv2 and UDv2 paper and repository for exhaustive loss definitions. We commit to explicitly documenting the loss configurations and other stage-2 training details in the final camera-ready version.
>
> DSFA operates exclusively as an encoder-side feature fusion module. To test whether the gain comes from generic finetuning, in table 4, we use the same training pipeline but replace the bokeh stack with repeated sharp frames at K=0 (w/o Bokeh). This causes a clear drop (δ1: 0.914→0.855), which rules out pure domain adaptation. These results prove the depth gains originate entirely from the calibrated defocus signal instead of the finetuning procedure.
>
> ## Response to Q
>
> We have expanded the ablation study of the Stage-2 DSFA presented in Table 4 below. The original "w/ BokehMe" configuration relies on GT sparse LiDAR depth, BokehMe, and DSFA. To provide a more comprehensive analysis, we introduce two additional variants. The first variant uses the predicted depth from DAv2 as the input for BokehMe. The second variant synthesizes the bokeh stack utilizing Flux-context. The updated evaluation results on the KITTI Eigen split under the Table 4 settings are summarized below.
>
> | Method | δ₁↑ | δ₂↑ | δ₃↑ | AbsRel↓ | SqRel↓ | RMSE↓ | RMSE(log)↓ | log10↓ | SiLog↓ |
> |-|-|-|-|-|-|-|-|-|-|
> | DAv2| 0.914 | 0.987 | 0.997 | 0.097 | 0.379 | 3.030 | 0.130 | 0.041 | 0.130 |
> | GT-Sparse-Depth + BokehMe + DSFA| 0.914 | 0.988 | 0.997 | 0.095 | 0.378 | 3.030 | 0.129 | 0.040 | 0.129 |
> | Pred-Dense-Depth + BokehMe + DSFA| 0.918 | 0.989 | 0.997 | 0.094 | 0.370 | 3.022 | 0.128 | 0.040 | 0.125 |
> | FLUX-Kontext + DSFA| 0.608 | 0.884 | 0.958 | 0.227 | 1.387 | 5.860 | 0.477 | 0.112 | 0.475 |
> | Ours (Stage-1 + DSFA)| 0.943 | 0.992 | 0.998 | 0.084 | 0.296 | 2.725 | 0.116 | 0.037 | 0.114 |
>
> These results support two clear conclusions. First, multiple bokeh patterns are helpful, but only when they are spatially usable. BokehMe with GT depth does not contradict our method. On KITTI, the ground truth is sparse LiDAR rather than dense per-pixel geometry. When such sparse depth is used for rendering, the resulting stack has unstable blur boundaries and weak spatial continuity. This limits how much DSFA can benefit from the added defocus signal. Once we replace sparse GT depth with a dense predicted depth map, performance improves. This shows that Stage-2 does benefit from multiple bokeh patterns, but it needs them to be dense and spatially coherent.
>
> Second, multiple bokeh patterns are still not sufficient without physical calibration. If DSFA were merely exploiting the visual style or artifacts of a generative backbone, then feeding it raw FLUX-Kontext bokeh rendering outputs should still help. In fact, it fails catastrophically. The result shows that Stage-2 does not benefit from arbitrary generated blur. It benefits from a stack whose blur changes are consistent across frames and aligned along a calibrated defocus axis. This is also exactly the design goal of our method, where Stage-1 produces a calibrated multi-strength bokeh stack and Stage-2 explicitly aggregates features along that axis. Our formulation and Proposition 4.1 are built on this calibrated sweep assumption.
>
> In short, the comparison does not suggest that DSFA is overfitting to FLUX-Kontext artifacts. It suggests a more specific conclusion: Stage-2 requires a bokeh stack that is dense, cross-frame consistent, and physically calibrated. BokehMe with sparse KITTI depth satisfies this only partially. Raw FLUX-Kontext satisfies none of it. Our Stage-1 is the only variant that satisfies all three, which explains why it delivers the largest gain. This interpretation is also consistent with prior bokeh rendering literature, where depth-guided rendering quality is known to depend strongly on boundary handling and occlusion consistency rather than depth input alone.

---

> > ### Author Rebuttal · Reviewer_MY8A · 2026-04-02
> >
> > Concerns have been addressed.

---

> > > ### Author Response · Authors · 2026-04-02
> > >
> > > We sincerely thank you for acknowledging our response and for raising your score to support our work. We are glad that your concerns have been fully addressed. Thank you again for your time and constructive review!

---

### Official Review · Reviewer_xxtu · 2026-03-13

**Soundness:** 3
**Presentation:** 4
**Significance:** 3
**Originality:** 3
**Overall Recommendation:** 5
**Confidence:** 4

**Summary:**

This paper proposes a two‑stage framework, BokehDepth, designed to enhance the prediction accuracy of existing depth foundation models. In the first stage, a bokeh stack is generated from a single sharp input image without requiring any depth information. The synthesized bokeh stack effectively introduces depth cues through defocus behavior. In the second stage, these defocus cues are exploited to improve the depth prediction accuracy. Extensive experiments demonstrate the effectiveness of the proposed method, showing significant improvements in both quantitative metrics and qualitative results.

**Compliance With Llm Reviewing Policy:**

Affirmed.

**Final Justification:**

Please see the Rebuttal Acknowledgement

**Key Questions For Authors:**

1. The term “sharp image” appears to refer to an “all‑in‑focus” input. However, real‑world images captured by consumer cameras often contain natural defocus cues. It would be helpful for the authors to discuss whether the method remains effective when such naturally defocused images are used as input, and whether any pre‑processing is required.
2. In Table 2, the performance improvement over UDv2 is considerably larger than that over DAv2. The authors are encouraged to analyze the underlying reasons.

**Limitations:**

yes

**Strengths And Weaknesses:**

Strengths:
1. The paper is well written, with a clearly articulated motivation, coherent narrative flow, and detailed explanations of the proposed method.
2. The proposed framework is technically sound, and the idea of leveraging additional depth cues generated by image‑generation models is compelling.
3. The paper provides extensive experimental evaluations and demonstrates substantial improvements over existing depth foundation models
4. The qualitative results are particularly impressive, showing significant improvements in fine‑grained depth details for distant or thin structures

Weaknesses:
1. Since the proposed framework introduces an additional model for bokeh‑stack generation as well as extra modules to process the stack, it would be beneficial to provide an analysis of parameter efficiency and inference speed.
2. Given that the first stage already produces bokeh stacks, one could directly estimate depth using off‑the‑shelf Depth‑from‑Defocus (DfD) models. It would strengthen the paper to (i) report the quality of bokeh‑stack generation from a DfD perspective and (ii) demonstrate that the proposed second‑stage refinement significantly outperforms existing DfD approaches.

---

> ### Author Rebuttal · Authors · 2026-03-31
>
> Thanks for your thoughtful review.
> ## Response to W1
>
> Our independent profiling reveals a clear decoupling of computational costs.
>
> | Component | Params | Latency (s/img) | Peak Mem (GiB) |
> |---|---:|---:|---:|
> | UniDepthV2 (baseline) | 353.83M | 0.045 | 2.48 |
> | UniDepthV2 + DSFA (precomputed stack) | 464.07M | 0.114 | 3.25 |
> | Stage-1 bokeh generation | 12.32B | 48.663 | 33.52 |
> | Full pipeline (Stage-1 + Stage-2) | 12.78B | 48.777 | 33.52 |
>
> Our core module for depth estimation, DSFA (Stage-2), introduces minimal overhead, while the computationally heavy Stage-1 bokeh generation operates entirely offline.
>
> DSFA adds just 110.24M parameters and 0.069 seconds of latency per image. This represents a marginal cost for the substantial performance gains demonstrated in Tables 2–4. Although Stage-1 accounts for 99.86% of the end-to-end latency, it serves strictly as an offline data-generation step akin to synthetic augmentation. Users can precompute, cache, and reuse the bokeh stacks across any downstream depth model. Once cached, the online inference cost becomes nearly identical to the unmodified baseline.
>
> We acknowledge the current full-scale diffusion backbone prohibits real-time Stage-1 execution. However, applying well-established one-step or few-step distillation techniques offers a direct optimization path. We will explicitly discuss this future trajectory in the final manuscript to clarify that the current latency is a solvable engineering challenge rather than a fundamental methodological flaw.
>
> ## Response to W2
>
> We compare BokehDepth against representative DfD methods on NYUv2 below.
>
> | Method | δ₁↑ | AbsRel↓ | RMS↓ |
> |---|---:|---:|---:|
> | Deep-Optics [1] | 0.930 | 0.087 | 0.433 |
> | DFF-DFV [2] | 0.967 | 0.445 | 0.232 |
> | 2HDED [3] | 0.914 | 0.029 | 0.244 |
> | DAIF-Net [4] | 0.950 | 0.170 | 0.325 |
> | SDNet [5] | 0.964 | **0.026** | 0.201 |
> | **Ours** | **0.978** | 0.039 | **0.043** |
>
> BokehDepth achieves the highest δ₁ and the lowest RMS by a large margin, outperforming all compared DfD methods.
>
> Importantly, this comparison is already tilted in favor of the DfD baselines. Every method listed above constructs its evaluation focal stack by rendering NYUv2 RGB-D pairs through a thin-lens or PSF forward model, meaning the stack implicitly encodes ground-truth depth. BokehDepth, by contrast, generates its bokeh stack from a single sharp image with no access to any depth map. Despite this strictly harder setting, our method still leads on the primary metrics. In real-world deployment where ground-truth depth is unavailable, the advantage would widen further.
>
> [1] Deep Optics for Monocular Depth Estimation and 3D Object Detection, ICCV 2019
>
> [2] Deep Depth from Focus with Differential Focus Volume, CVPR 2022
>
> [3] Depth Estimation and Image Restoration by Deep Learning from Defocused Images, IEEE Trans. Comput. Imaging 2023
>
> [4] Fully Self-Supervised Depth Estimation from Defocus Clue, CVPR 2023
>
> [5] Depth Estimation Based on 3D Gaussian Splatting Siamese Defocus, ICRA 2025
>
> ## Response to Q1
>
> We sincerely appreciate the reviewer for highlighting this practically valuable research direction. Utilizing sharp input is an established community convention rather than a unique assumption of BokehDepth. Standard benchmarks (e.g., KITTI, NYUv2, IBims-1) and all evaluated baselines rely on all-in-focus images because the core objective is extracting 3D depth from 2D RGB information..
>
> Paradoxically, BokehDepth is intrinsically better equipped to handle naturally defocused inputs than these standard methods. While conventional models treat blur as degradation noise, our DSFA module is explicitly optimized to extract geometry from structured defocus variations. However, systematically validating this advantage requires a currently non-existent benchmark containing paired naturally defocused images and ground-truth depth. We therefore leave the construction of such a dataset and the thorough exploration of this promising setting to future work.
>
> ## Response to Q2
>
> We attribute the difference to architectural compatibility.  UniDepthV2 explicitly conditions depth features on a learned camera representation and disentangles camera geometry from scene depth in its output space. This design provides a natural entry point for physically grounded image-formation cues. The calibrated bokeh signal from DSFA aligns directly with UniDepthV2's geometric conditioning pathway, allowing the model to absorb defocus variation more effectively. Depth Anything V2, by contrast, follows a pure appearance-to-depth regression paradigm trained on massive and diverse data.. Crucially, DSFA improves both backbones across all benchmarks in Table 2, confirming that the module is architecture-agnostic. The difference in gain magnitude reflects how much complementary geometric capacity each backbone exposes, not a limitation of the method itself.

---

> > ### Author Rebuttal · Reviewer_xxtu · 2026-04-02
> >
> > The reviewer thanks the ahtors for their detailed response and addtional experimental results. After reading the response, my concerns are fully addressed, therefore I decide to keep my positive rating.

---

> > > ### Author Response · Authors · 2026-04-02
> > >
> > > We sincerely thank you for acknowledging our response and for your continued support of our work. We are glad that your concerns have been fully addressed. Thank you again for your time and constructive review!

---

### Official Review · Reviewer_AFQN · 2026-03-13

**Soundness:** 3
**Presentation:** 3
**Significance:** 3
**Originality:** 3
**Overall Recommendation:** 4
**Confidence:** 4

**Summary:**

This paper proposes BokehDepth, a two-stage framework that leverages synthetic defocus blur as a supervision-free geometric cue for monocular metric depth estimation (MMDE). Stage-1 builds a controllable bokeh generator on FLUX-Kontext, conditioned on a single thin-lens-derived scalar K, to produce calibrated multi-strength bokeh stacks from a single sharp image without any depth input. Stage-2 introduces a Divided Space Focus Attention (DSFA) module that is inserted into the encoder of existing discriminative depth models (DAv2, UniDepthV2); DSFA first performs spatial attention within each bokeh frame, then cross-frame focus attention along the blur-strength axis, injecting defocus cues via FiLM modulation while leaving the decoder and metric head unchanged. Experiments show consistent improvements over DAv2 and UDv2 on six zero-shot benchmarks and NYUv2, alongside state-of-the-art bokeh rendering quality on EBB! Val200 and SYNTHEBOKEH300.

**Compliance With Llm Reviewing Policy:**

Affirmed.

**Key Questions For Authors:**

1. How does the number of bokeh stack frames N and the range of K values affect Stage-2 depth accuracy? Even a small sweep (N $\in$ {1, 2, 3, 5}, K ranges of [5,15] vs [10,30] vs [20,50]) would clarify this key design axis and its interaction with inference cost.

2. What happens when Stage-1 bokeh is replaced by BokehMe renderings driven by the base model's own (noisy) depth prediction rather than ground-truth depth? This would isolate Stage-1's contribution from DSFA's.

**Limitations:**

Yes

**Strengths And Weaknesses:**

Strengths:

S1. The decoupling of bokeh synthesis from depth prediction is well-motivated. Proposition 4.1 provides rigorous theoretical grounding: under the thin-lens model, regressing calibrated bokeh radii against K yields an unbiased, consistent inverse-depth estimator, which directly justifies the DSFA design.

S2. DSFA is genuinely plug-and-play — only the encoder is modified while the DPT decoder and metric head remain untouched (Eq. 5-7). Tables 2-3 confirm this generality across DAv2 and UDv2.

S3. Evaluation covers both tasks thoroughly: four bokeh baselines on real and synthetic benchmarks (Table 1), six zero-shot depth datasets plus NYUv2 in-domain (Tables 2-3). The depth-stratified MAE in Table 7 further shows gains concentrate at larger depths, consistent with the physics of defocus cues.

S4. The DSFA ablation (Table 4) cleanly isolates each design choice (Focus/Space branches, FiLM, bokeh source) with clear conclusions.

Weaknesses:

W1. The paper fixes the bokeh stack size at N=3 frames with K $\in$ [10, 30] but provides no ablation on either hyperparameter. N directly governs the computation-vs-information trade-off: is N=1 (a single defocused frame) already sufficient, or does N=5 yield further gains? Proposition 4.1 shows the OLS variance decays as $1/\sum K_i^2$ (Eq. 36), which provides a principled lens for this analysis — more frames with larger K spread should reduce estimation variance. Similarly, the K range determines the effective depth-of-field probed by the stack; a narrow range may miss shallow-focus cues while an overly wide range may introduce excessive blur artifacts. This is a critical pipeline-level design choice that lacks both experimental and theoretical justification.

W2. Table 4 compares Stage-1 against BokehMe driven by ground-truth depth, but the practically relevant ablation is missing: using the base model's own predicted depth to drive a classical renderer (e.g., BokehMe with DAv2's initial depth output) as a cheap Stage-1 substitute. If this simple alternative already provides useful defocus cues to DSFA, the marginal value of the expensive generative Stage-1 would need re-evaluation. Without this experiment, the necessity of Stage-1's large diffusion-based generator remains an open question.

---

> ### Author Rebuttal · Authors · 2026-03-31
>
> Thanks for your thoughtful review.
>
> ## Response to W1&Q1.
>
> We have now added a direct sweep over both hyperparameters under the same Table 4 setting on the KITTI Eigen split.
>
>
> | Variant | δ₁↑ | δ₂↑ | δ₃↑ | AbsRel↓ | SqRel↓ | RMSE↓ | RMSE(log)↓ | log10↓ | SiLog↓ |
> |---|---:|---:|---:|---:|---:|---:|---:|---:|---:|
> | N=1, K∈[10,20] | 0.91827 | 0.98731 | 0.99746 | 0.09497 | 0.37013 | 3.02922 | 0.12906 | 0.04009 | 0.12520 |
> | N=1, K∈[20,30] | 0.91794 | 0.98692 | 0.99700 | 0.09507 | 0.37000 | 3.02620 | 0.12905 | 0.04001 | 0.12486 |
> | N=2, K∈[10,20] | 0.92821 | 0.98776 | 0.99715 | 0.08599 | 0.34170 | 2.88064 | 0.12003 | 0.03749 | 0.12496 |
> | N=2, K∈[20,30] | 0.92832 | 0.98777 | 0.99715 | 0.08591 | 0.34077 | 2.87534 | 0.11995 | 0.03744 | 0.12487 |
> | N=2, K∈[10,30] | 0.93161 | 0.98880 | 0.99746 | 0.08542 | 0.33210 | 2.84620 | 0.11892 | 0.03744 | 0.12235 |
> | **N=3, K∈[10,30]** | 0.94302 | 0.99233 | 0.99843 | 0.08420 | 0.29619 | 2.72478 | 0.11633 | 0.03742 | 0.11416 |
> | N=5, K∈[10,30] | 0.94331 | 0.99258 | 0.99847 | 0.08382 | 0.28974 | 2.69135 | 0.11558 | 0.03696 | 0.11298 |
>
> These numbers support two clear conclusions. First, a single defocused frame is not sufficient. Moving from $N=1$ to $N=2$ gives a large gain, and $N=3$ improves further. By contrast, increasing from $N=3$ to $N=5$ gives only a small additional improvement. Second, for the $K$ range, the effective spread matters more than the absolute location. Under fixed small $N$, $[10,20]$ and $[20,30]$ perform almost identically, while the wider range $[10,30]$ consistently performs better. This means that DSFA benefits primarily from a sufficiently spread calibrated defocus sweep, not from simply shifting the interval to larger $K$ values.
>
> This is exactly what Proposition 4.1 proves. Our theory models depth recovery as the slope of a per-pixel regression on $K$. Increasing the number of observations and increasing the spread of $K$ both improve the conditioning of this regression. The empirical sweep follows this trend closely.
>
> We also measured the cost of increasing $N$ for DAv2 based Bokehdepth.
>
> | N | Stage-1 mean (s) | Stage-2 mean (s) | Full mean (s) | Stage-1 mem (GB) | Stage-2 mem (GB) |
> |---|-----------------:|-----------------:|--------------:|-----------------:|-----------------:|
> | 1 |            16.76 |             0.08 |         16.84 |            33.08 |            26.20 |
> | 2 |            32.54 |             0.10 |         32.64 |            33.70 |            26.23 |
> | 3 |            48.66 |             0.12 |         48.78 |            34.32 |            26.33 |
> | 4 |            64.46 |             0.14 |         64.60 |            34.94 |            26.53 |
> | 5 |            79.50 |             0.16 |         79.66 |            35.56 |            26.80 |
>
> The runtime is dominated by Stage-1 and scales almost linearly with the stack size.  Over the same range, the accuracy gain largely saturates after $N=3$.  Our added sweep answers this directly and justifies $N=3,\ K\in[10,30]$ as the best trade-off in the tested regime.
>
> ## Response to W2&Q2
>
> The table below includes your suggested variant (DAv2-Pred-Depth + BokehMe) in the setting of Table 4. **Full results with all metrics and analysis are provided in our response to Reviewer MY8A (Response to Q).**
>
> | Method | δ₁↑ | AbsRel↓ |
> |---|---|---|
> | DAv2 (baseline) | 0.914 | 0.097 |
> | Pred-Depth + BokehMe + DSFA | 0.918 | 0.094 |
> | **Ours (Stage-1 + DSFA)** | **0.943** | **0.084** |
>
> Stage-1 drastically outperforms the BokehMe substitute. This substantial gap occurs because BokehMe suffers from a circular dependency on noisy initial depth predictions, whereas our generative Stage-1 completely breaks this error loop by synthesizing calibrated defocus cues independently of any depth prior.

---

> > ### Author Rebuttal · Reviewer_AFQN · 2026-04-07
> >
> > The authors have thoroughly addressed both of my concerns. The N/K hyperparameter sweep (W1) provides clear empirical justification aligned with Proposition 4.1, and the Pred-Depth+BokehMe comparison (W2) convincingly demonstrates the necessity of the generative Stage-1. I also find the expanded Table 4 addressing Reviewer MY8A's overfitting concern particularly compelling. I am satisfied with the rebuttal and maintain my original score.

---

> > > ### Author Response · Authors · 2026-04-07
> > >
> > > We sincerely thank you for acknowledging our response and for your continued support of our work. We are glad that your concerns have been fully addressed. Thank you again for your time and constructive review!

---

### Decision · Program_Chairs · 2026-04-30

**Decision:**

Accept (regular)

**Comment:**

The paper received consistent positive recommendations from four reviewers, including three weak accepts and one accept. The reviewers acknowledged the novelty of the proposed model, the solid experimental evaluation, and the high-quality presentation of the paper. AC agrees with the comments of the reviewers and recommends an acceptance for this submission.